# Break down the decentralization-security-privacy trilemma in management of distributed energy systems

Qinghan Sun [1], Huan Ma[1], Tian Zhao[1,2], Yonglin Xin[1] & Qun Chen [1,2]✉

Distributed energy systems encompass a diverse range of generation and storage solutions on the user side, where decentralized management schemes to maximize the overall social welfare are preferred considering their dispersed ownership. However, either security or privacy problems occur in recently proposed schemes. Here we report a decentralized framework leveraging the strengths of blockchain and parallelizable mathematical algorithms to overcome these potential drawbacks. The system owners bid cost functions and operating constraints through masked but coupled management subproblems, which are redistributed by the blockchain to be verifiably solved by competent peers. Such processes are iteratively executed as decisions and shadow prices are exchanged among participants, until an equilibrium is reached. The interactive framework ensures decentralized, privacy-preserving, and secure management of multiple energy sources, and reduces the total cost by 3.0 ~ 7.5% in the test system. Our results benefit the energy prosumers and promote a more active and competitive power grid.

Distributed Energy Systems(DESs) are promising user-side solutions to accomplishing carbon neutrality[1] and handling climate challenges[2]. This concept exploits the potential of energy customers to employ local devices such as photovoltaic(PV) panels[3], micro turbines[4], and batteries[5] to provide flexible energy services[6]. In this context, conventional energy consumers enter the business of energy production and provision, serving as energy prosumers[7]. This role transition implies a mutually beneficial situation, where individuals can pursue personal interests by adjusting operation strategies, positively impacting social welfare.

However, the key problem is coordinating these DES entities to maximize collective benefits. Typically, when DESs deviate from design regimes to meet diverse and volatile demands, the energy utilization efficiency may be reduced[8]. The cooperation through scheduled energy transport and trading among DESs can attenuate such effect, but their respective owners may not be willing to transfer the operation authority to a third party to realize overall coordination.

Decentralized management of multiple DESs is a possible solution. Specifically, the decision is not made by a single party with a superior position. Everyone's perception is incorporated into decision-making on the expected operation status of the DESs for a certain target, such as minimization of total energy cost. The management problem is usually modeled as mathematical programming[9] and solved by everyone cooperatively. When designing decentralized schemes, functional requirements, namely, feasibility and optimality of an operation plan considering different techno-economic objectives and constraints, are important considerations. Moreover, non-functional aspects, such as usability for lightweight participants, computational performance, security[10], and degree of decentralization[11] are also major concerns.

Blockchain is a powerful solution to realize secure decentralized management. Through smart contracts, the publicly-agreed functionalities are encoded on the blockchain and executed by miners[12], during which process a consensus must be formed among participants. Proof of Work(PoW)[13], Proof of Stake(PoS)[14], and practical Byzantine Fault

[1]Key Laboratory for Thermal Science and Power Engineering of Ministry of Education, Department of Engineering Mechanics, Tsinghua University, Beijing, China. [2]School of Energy Storage Science and Engineering, North China University of Technology, Beijing, China. ✉e-mail: chenqun@tsinghua.edu.cn

Tolerance(pBFT)[15] are some widely-used consensus schemes. To enhance the performance in applications to energy sectors, protocols such as Proof of Clearance(PoC)[16] and Proof of Solution(PoSo)[17] are also proposed. In the current blockchain-based paradigm, the optimization puzzle of connected DESs is disclosed to all people. Then the puzzle is solved by authenticated participants[18] and verified by endorsing peers[16,17,19] to avoid malicious manipulations. In other words, decentralization in blockchain is realized through open but integrated decisions. We argue, however, that such schemes require aggregating and exposing private information, such as load patterns, user behaviors[20], and information on local generation devices on the chain. Interested ones may identify DES owners' schedules and activities and benefit from these personal data[21,22]. Critical power network topology of multiple DESs is also leaked online, exacerbating worries about attacks[23].

Parallelizable algorithm(PA) for mathematical optimization is another popular decentralized implementation[24,25] but differs in the idea behind it. PA-based design enables local computation to maximize the degree of decentralization and confidentiality[26]. Through mathematical algorithms such as distributed dual ascent[27] and the Consensus+Innovation method[28], the integrated optimization problem is decomposed and every DES collects information from neighbors and solves local subproblems iteratively to reach a consensus. Despite these privacy-preserving advantages, solving mathematical programming problems repeatedly is a computationally intensive task[29]. The complex localized calculation raises the threshold for lightweight participants and enlarges the size of the trusted computing base(TCB)[30]. In other words, more room is left for active manipulations during the decision process. The participants may cheat each other for unjust personal profits[31].

Blockchain and PA focus on different concerns of decentralization and thus indicate respective limitations on privacy or security. Briefly speaking, privacy and secured decisions are two somewhat contradictory goals in the context of decentralization, as illustrated in Fig. 1. The path toward privacy in decentralized management emphasizes high-degree distributed computing. Such reliance on localization might be more susceptible to modifications from the DES owners. The other path toward security asks for public confirmation and voting, which leads to privacy disclosure. So far, several studies have realized the security risks in PA and introduced blockchain to enhance the integrity of the PA-based management process[31-36]. They use blockchain to aggregate the decision results provided by DES participants according to different PAs and endorse its consistency, i.e., a DES participant cannot fool others with different results and all participants work on the same iteration status. Such data aggregation in plaintext compromises the privacy-preserving feature of PA. In some design, price signals may be the only exchanged information to protect privacy[36-38]. However, these methods still lack support for DESs with diverse energy supply characteristics, and the power transmission process and related physical constraints are not well considered. Apart from this, dishonest individual participants may still deliberately disrupt the convergence process to hazard security or cheat by providing consistent but false information. To solve the trilemma of decentralized coordination, security and privacy, not only the data transferred on the public blockchain should be obfuscated and verifiable, but also the off-chain calculations should be tamper-proof.

In this work, we propose a mechanism to address the privacy and security concerns in the decentralized management of DESs. The basic philosophy follows state-of-the-art decentralization solutions, where optimization subproblems are formulated for DES participants and iteratively solved to reach a consensus. To protect the DESs' cost functions and equipment capacities, the subproblems are encrypted before being sent to the blockchain, which can then be efficiently solved off-chain by competent computation parties and verified on-chain by miners. To avoid the adversarial behavior of DES owners, we reduce their local computation to fundamental arithmetic operations of matrices so that hardware-backed edge devices using Trusted Execution Environment(TEE)[39] are deployed to avoid disobedience to protocols. On the data of a real 10kV distribution grid in Yingkou City, China, we compare the proposed method with other decentralization approaches considering potential manipulative actions. We also set up a real communication network to demonstrate the feasibility and effectiveness of the framework.

## Results
### Formulate, encrypt, and distribute the optimization tasks
We focus on making day-ahead operating decisions. In the following, we treat each DES as managed by an independent social entity. Multiple DESs are connected through transmission lines. We aim to guide the DES owners to control the energy flow, or more concretely, the voltage and power set points of the DESs and transmission lines, to minimize the total bid cost.

The proposed framework to mend the gap between security and privacy is illustrated in Fig. 2. In general, there are three kinds of entities participating in the proposed scheme, which are DES owners, blockchain miners, and computation parties. The three identities may overlap. Besides, specially designed edge computing equipment or smart meters supported by TEE technology, e.g., Intel SGX[40] or ARM TrustZone[41], are deployed locally at each DES. TEE is a segregated area of CPU and memory to ensure the integrity and privacy of the code and data. It serves as the interface between raw bid data from DES owners and pre-defined scheme procedures. The programs run in the edge devices are supervised by corresponding DES owners but also well protected from their modifications.

Before the decentralized decision-making begins, the DES owners should prove that they have enough currency for both calculation service fee and electricity bills[42]. The currency can be arbitrary cryptocurrencies that natively protect transaction details such as Monero[43] or Zcash[44]. Our framework does not limit to one specific coin for settlement because transactions are bilateral and can be selected depending on the willingness of each couple of entities. Besides, the blockchain miners should attest the DES owners, their TEE-based edge devices, and the computation parties, and then record the attestation results on the blockchain.

The main part is the interactive decision-making process among the participants. In Step 1 of Fig. 2, the DES owners first provide

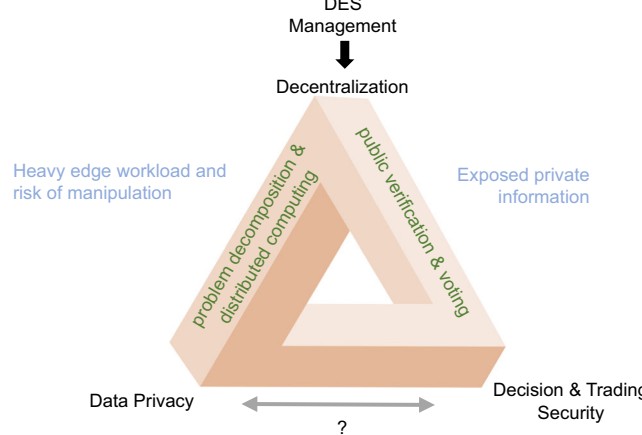

**Fig. 1 | Trilemma in management of DESs: decentralization, privacy, and security.** Decentralization means that the final decision is not made by a single party. Researchers may use PA or blockchain to realize decentralization. The former requires everyone to solve their own optimization subproblems and focuses on data privacy. The latter usually lets the blockchain be the "centralized" decision maker and emphasizes security. A simple combination of PA and blockchain may compromise both security and privacy.

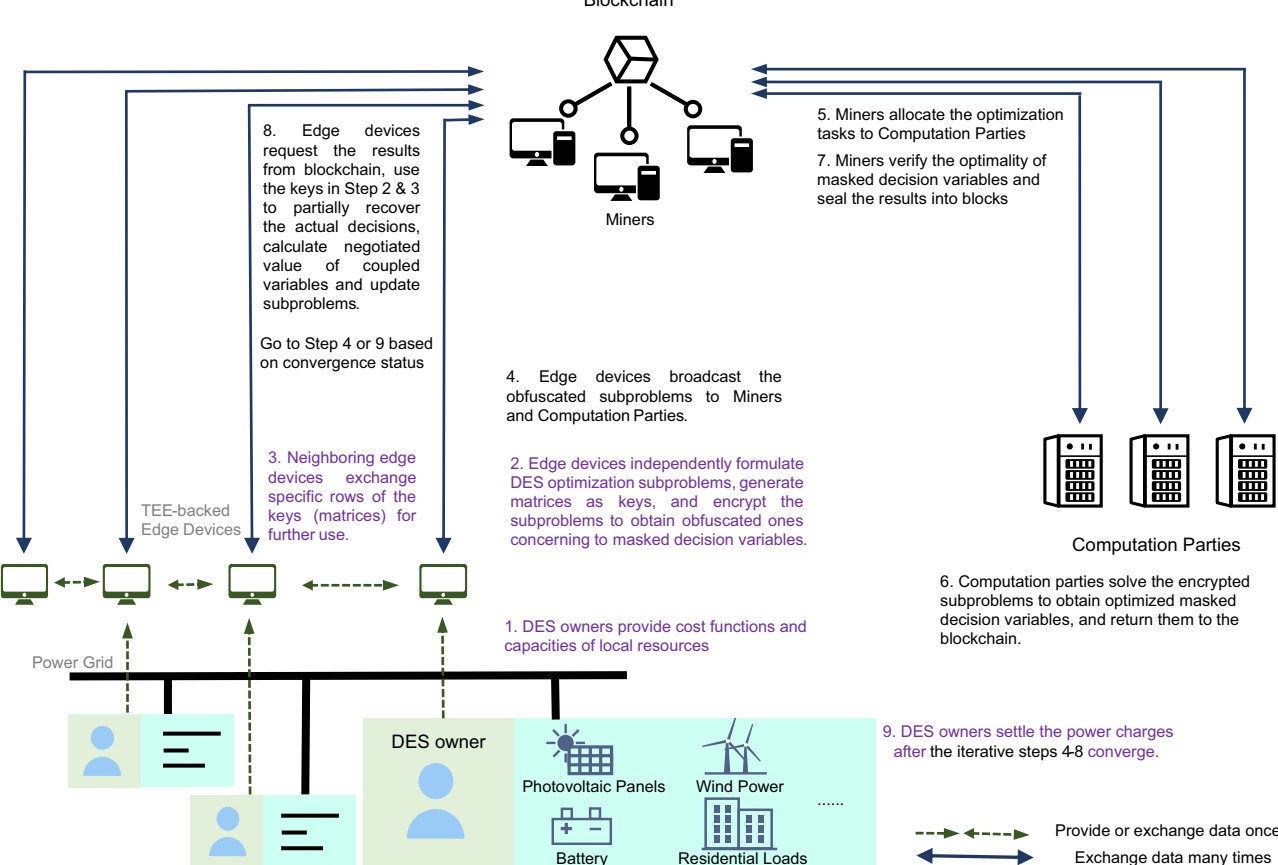

**Fig. 2 | Overview of the proposed scheme.** A total of 9 operation steps performed by different parties are presented, where steps 4–8 are executed iteratively. The following trust assumptions are made: i. DES owners are only data providers. ii. Embedded edge devices with TEE functionalities are deployed for each DES for edge data processing, aggregation, and protection. They are remotely attested by the blockchain miners. iii. Miners are maintainers of the blockchain while computation parties are professional computation service providers offering decision-making support for the DES owners. We allow arbitrary malicious behavior for them and assume a majority of miners(more than 2/3) and at least one computation party still follow the protocol completely.

information on generation sources or batteries as bids, which involve sensitive parameters such as capacity and quadratic generation cost functions, to their edge computing devices. Then, based on parallelizable optimization methods, in particular, the Alternating Direction Method of Multipliers(ADMM)[45] in this paper, the edge devices validate DES owners' input and formulate optimization subproblems for each DES independently. The decision variables of the subproblems are set points of local generation resources. The objectives include both the owners' operating costs and augmented-lagrangian-based penalties, which are given as,

$$\min_{\mathbf{x}_i = [\mathbf{x}_{d,i}, \mathbf{x}_{c,i,j} \forall j \in \mathcal{N}_i]} \underbrace{\left(\frac{1}{2}\mathbf{x}_{d,i}^\top \mathbf{H}_{d,i}\mathbf{x}_{d,i} + \mathbf{c}_{d,i}^\top \mathbf{x}_{d,i}\right)}_{\text{bid operation costs}}$$
$$+ \sum_{j \in \mathcal{N}_i}\underbrace{\left[\boldsymbol{\lambda}_{i,j}^\top(\mathbf{x}_{c,i,j} - \mathbf{e}_{c,i,j}) + \frac{1}{2}\|\mathbf{x}_{c,i,j} - \mathbf{e}_{c,i,j}\|_\Theta^2\right]}_{\text{augmented-Lagrangian-based penalties}} \quad (1)$$

where $\mathbf{x}_{d,i}$ denotes the decision variables of distributed sources owned by DES agent $i$ and $\mathbf{x}_{c,i,j}$ denotes $i$'s decision variables that couple with its neighbor $j$, such as active and reactive power transmission. $\mathbf{e}_{c,i,j}$ is the negotiated average value of $\mathbf{x}_{c,i,j}$ and $\mathbf{x}_{c,j,i}$. $\boldsymbol{\lambda}_{i,j}$ is the Lagrangian multipliers of coupling constraints. The penalties will be modified and designed as the transaction cost in the settlement stage afterward. If nothing is provided by the DES agent, still a problem without $\mathbf{x}_{d,i}$ will be generated by the power grid owner instead.

In Step 2, the subproblems will be further encrypted by a group of matrices $\mathbf{N}_i$, $\mathbf{R}_i$ and $\mathbf{x}_i^0$ (see Methods for a detailed explanation) to obfuscate the bid parameters. The subproblem after obfuscation is a new one concerning a set of masked decision variables $\mathbf{y}_i$. The matrices for encryption are kept by DES owners as keys. Later after the computation parties solve the sticky subproblems and obtain masked solutions, different rows of the matrices can be used to recover the different elements of the real optimal decision variables $\mathbf{x}_i^*$. In Step 3, the rows of $\mathbf{N}_i$, $\mathbf{R}_i$ and $\mathbf{x}_i^0$ corresponding to $\mathbf{x}_{c,i,j}(\mathbf{x}_{c,j,i})$ will be exchanged between neighboring DES $i$ and $j$, so that they can decrypt the coupling decision variables and calculate the negotiated value of $\mathbf{e}_{c,i,j}$.

Once those preparative steps are finished, the obfuscated subproblems are broadcast to the miners and computation parties(Step 4). Meanwhile, the miners maintain on the blockchain a task allocation schedule, which is generated through pre-defined functions according to solution performances of the computation parties and hash values of previous blocks. The computation parties solve the subproblems accordingly and re-broadcast the solutions online(Steps 5 & 6). The miners verify the optimality of solutions and package them into a block through efficient consensus algorithms(Step 7). Once upon a new block is confirmed, the edge devices recover optimization solutions and partial results of their neighbors' subproblems to calculate $\mathbf{e}_{c,i,j}$ and update consensus penalties. Finally, a new iteration epoch begins.

The iteration terminates when sufficient miners have agreed on convergence, i.e., when $\mathbf{x}_{c,i,j}$ and $\mathbf{e}_{c,i,j}$ and their values between two consecutive iterations are close enough. The detailed formulation and encryption process of optimization subproblems, the consensus

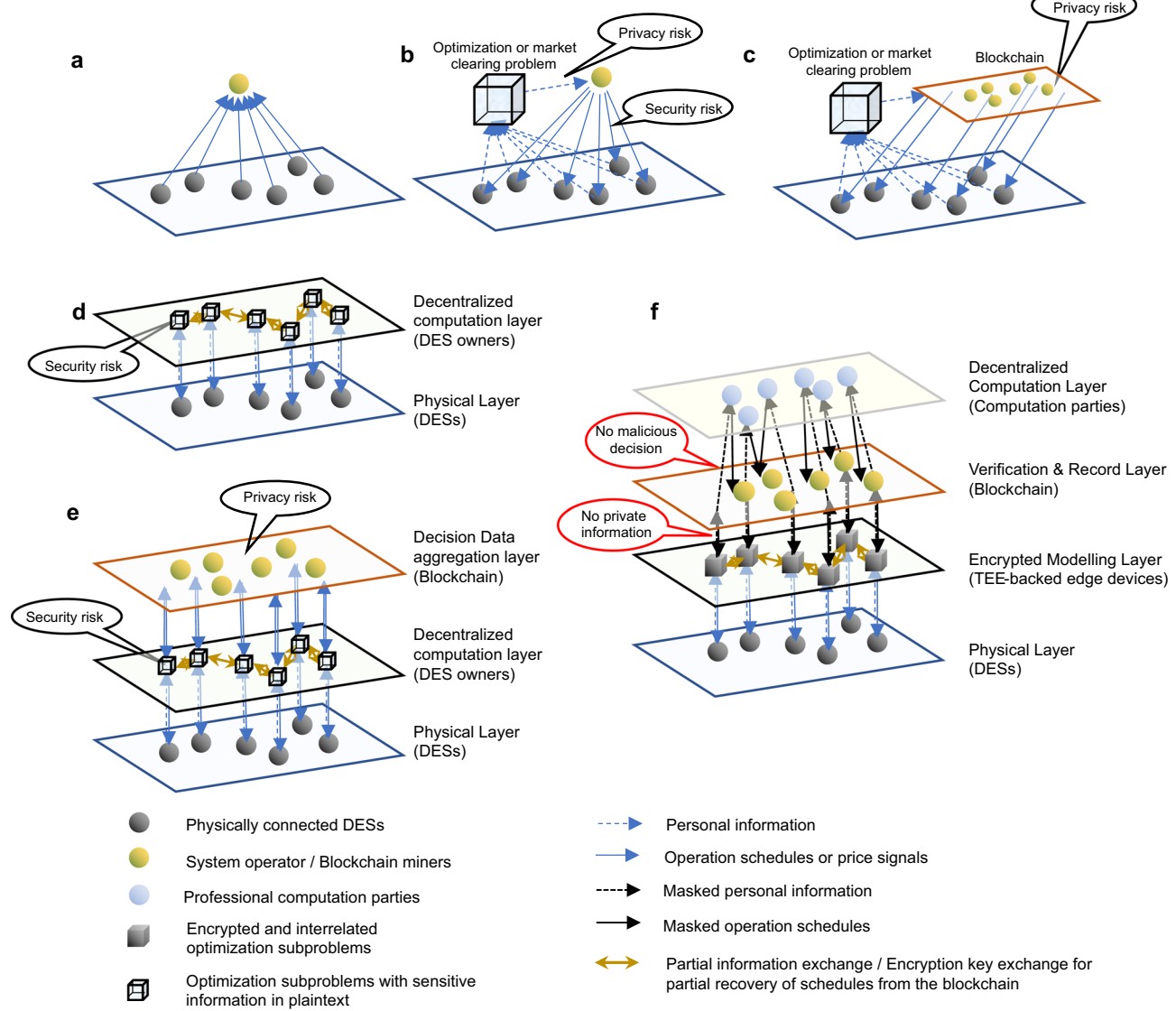

**Fig. 3 | Comparisons between different operation paradigms. a** The DESs operate in a non-cooperative manner. They tell the system operator the amount of electricity that they want to purchase, and the operator fulfills all requests. **b** The DESs cooperate through a centralized dispatch. They submit information of their equipments to the operator and wait for its operation schedule. **c** Blockchain is employed to replace the centralized operator. **d** PA is used for decentralization. DES owners solve local optimization subproblems, communicate with each other, and iterate to convergence. **e** Existing works that directly combine PA and

blockchain. Blockchain is used to ensure consistency in the optimization status. However, dishonest DES participants can still provide falsified operation schedules and compromise security. Privacy problems remain. **f** The proposed decentralized cooperation paradigm, where the centralized optimization problem is locally decomposed into subproblems and securely encrypted by edge devices to mask sensitive information. With blockchain and computation parties, they are verifiably solved to ensure robust convergence to optimality.

protocol used by miners, and the subproblem update process used in this work are thoroughly described in the Methods section and Supplementary Notes 1 and 2.

After the completion of the iteration and the implementation of the agreed-upon operation plan, the settlement stage begins (Step 9). During this stage, the miners and workers are duly compensated for their contribution to the decision-making process. More importantly, the DES owners pay their energy charges to each other so that they are willing to continuously participate in the decentralized management. The commonly used locational marginal pricing(LMP) sets different prices for different nodes of the power system. However, it does not identify specific payees for buyers or payers for sellers and is incompatible with decentralization. In this work, we employ a neighbor-to-neighbor payment system considering the invisibility of the interconnected system as a whole. The multipliers, also known as shadow

prices, of consensus constraints between neighbors, are used for pricing. Each DES participant $i$ is required to directly compensate its neighbors $j$ $\lambda_{i,j}^{\top} \mathbf{x}_{c,i,j}$. The resulting prices $\lambda_{i,j}$ vary across neighboring DESs and emerge as incidental benefits from the iterative solutions. They mirror the concept of LMP but are interpreted from a decentralized perspective. This payment structure ensures both efficiency and positive outcomes for all DES participants. For a more detailed discussion, please refer to the Methods section.

In Fig. 3, we compare different operation paradigms. Fig. 3a sketches the conventional non-cooperative scenario, while in Fig. 3b the operator should thoroughly consider the operation constraints of all DESs, minimize the cost and use the LMP for settlement. In the blockchain-only paradigm shown in Fig. 3c, a blockchain comprising several miners replaces the role of a single operator to avoid manipulation. In the PA-only paradigm shown in Fig. 3d, the DES owners may

manipulate the subproblem solutions to disrupt the convergence process and pursue unjust personal profits. In popular works that directly combine PA and blockchain shown in Fig. 3e, the blockchain ensures that the global decision status is consistent and tamper-proof. However, the DES owners, which are interested parties, are still left with authority to locally solve the subproblems. This leaves a loophole and secured convergence to optimality may not be guaranteed by blockchain. Besides, the data aggregation on blockchain compromises privacy. In comparison, our decentralization method is illustrated in Fig. 3f. Herein, we divide different participants into different layers with different functions. On the Encrypted Modeling Layer, sensitive data are broken into encrypted slices and only the DES owner and its neighbors with correct keys can fully or partially recover the original decision variable. On the Decentralized Computation Layer, participants can be arbitrary anonymous computation sources and they can share the computation power freely. In the Verification & Record Layer, trusted miners check the feasibility and optimality of subproblem solutions. Such a framework ensures a secure and privacy-preserving decentralized management.

## Instantiation: computation performances and economic benefits

To evaluate the effectiveness of the proposed framework, we utilize data from a real 10 kV power distribution grid in Yingkou City, Liaoning Province, China, as shown in Fig. 4a, b. This distribution grid is modeled as a 60-bus system with 1 transformer substation, 48 nodes with industrial and commercial customers, and 11 intermediate nodes with no loads. 21 buses are equipped with self-provided gas-fired generators, PV panels, distributed wind farms, or energy storage devices. Detailed explanation of the test distribution system as well as configuration and modeling of the DESs can be found in Supplementary Note 3. Without loss of generality, we suppose there are in total 60 independent DESs in the test system. The dispatch and transaction time interval is 15 min and there are 96 decision points in one day.

At present, the system is operated in a non-cooperative manner. All of the DES participants can freely adjust local devices and directly deal with the grid operator according to the time-of-use(TOU) price table and feed-in tariff. As a result, the average net load of this distribution grid during the week is 2.13 MW, and the total energy cost is shown in Table 1.

The proposed decentralized optimization framework is continuously applied to the system for the same week. That is, the iterative process to reach a consensus is repeated 7 times, each covering the operation decisions and transactions in one day. Implementation details can be found in the "Methods" section. The iteration curves of the total operating cost of all DES participants and the residuals are presented in Fig. 4c, d. The proposed decentralized framework reaches the residual threshold of $10^{-3}$ with about 735 iterations on average during the test week and the total iteration number is 5149. The near-zero primal and dual residuals respectively indicate that the neighboring DESs have agreed on each other's voltage and active and reactive power transmission through lines and that the DESs' decisions on operation status no longer change. The averaged maximum computation time on each encrypted subproblem in each iteration is 0.15 s, which is increased by about 50% compared to optimization on the original subproblem(0.10 s). This is mainly due to the loss of the sparse features of the original subproblem during the encryption process, but is in general acceptable. Since we offload the subproblem solution process to third parties, the edge devices owned by DESs are only required to update some parameters of the subproblems, and the maximum time consumption for this update is 0.001 s, which is almost negligible.

The final energy cost of the decentralized optimization and theoretically optimal results are also presented in Table 1 for comparison. With decentralized management, the DESs are united to transact with

the upper-level grid, and the daily energy cost is reduced by 3.0 - 7.5%. The results obtained by decentralized management are quite close to those obtained through an ideal centralized dispatcher. The centralized optimization is accurate and efficient as it assembles all of the DES data required, but faces authority and privacy problems in practice. Besides, the statistics of the test week presented in Table 1 illustrate that, the benefits of cost reduction are shared among the DESs as they are offered with more power generation alternatives at affordable prices. Supplementary Note 4 supplements some detailed operation status of the DESs in the non-cooperative and decentralized management case.

## Analysis of privacy protection and adversarial behaviors

Here, we first show that the decision-making process protects each individual's privacy. Take DES 7's owner as an example, who bids operational information of 2 distributed energy generation and storage devices. To model DES 7's operation status as well as the power flow status, a decision vector of length 1536 is organized by the corresponding edge device, and its final value is shown in Fig. 5a. Different slices of the vector's value reveal the optimal decision details of DES 2, as highlighted by the rectangle markers with numbers. From this information, the behavior patterns of energy usage of DES 7 can be easily inferred. In our framework, a masked decision vector with a lower dimension of 716 is employed as an alternative to be exposed to the miners and the workers, whose values are presented in Fig.5b. The miners and workers can only receive optimization objectives, constraints, and values of the masked variables, while the true information of DES 7 is concealed. The map between the original decision vector and the masked vector is only known to DES 7's owner, and unauthorized parties cannot glimpse this privacy. Besides, the eigenvalues of the encrypted quadratic cost function matrix $\mathbf{H}'_i$ are also disrupted, as shown in Fig. 5c. The true cost functions of the DESs are protected as well.

We then examine potential adversarial or strategic behaviors exhibited by the participants. For the computation parties and blockchain miners, their access is strictly limited to the encrypted DES subproblems stored on the blockchain. Throughout the decision-making process, the solutions provided by the computation parties are under constant supervision by both the blockchain miners and DES owners. If they slow down the solution process, either with intention or due to solution inefficiencies, they may be allocated less computation tasks or none as a penalty. Consequently, they cannot manipulate the decision process and results or compromise the privacy of DESs. As for the blockchain miners, our system relies on the pBFT scheme, which can tolerate up to one-third of the miners engaging in arbitrary malicious behavior. The pBFT leader should completely follow the process of scheduling computation assignments and verifying the solution results. If the current pBFT leader attempts to cheat, a new leader will be promptly elected to take place.

To further illustrate the effectiveness of the proposed method in reaching secure and privacy-preserving decentralized management of DESs, we compare the proposed method with existing decentralized techniques considering the following manipulation behavior,

a. (dishonest DES agent) The DES owners may be required to perform some local calculations and communicate with their neighbors. The dishonest ones may provide wrong or inconsistent calculation results for more profits. They are rational, and thus will not try to ruin the decentralized management process.

b. (manipulative third-party decision-maker) Some third-party participants other than DES owners, may be required to optimize the operation of the DESs. However, the manipulative decision-maker colludes with some DES and tries to make tendentious operation plans or cheat on the energy prices.

c. (irrational malicious participant) The irrational participant wants to prevent the management process from reaching

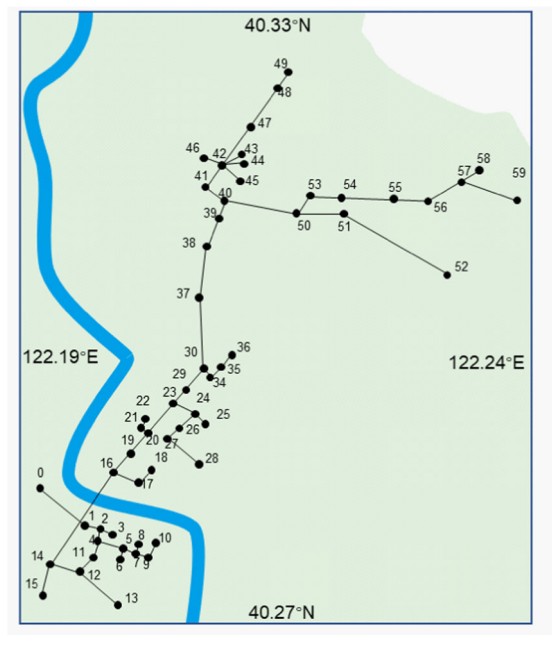

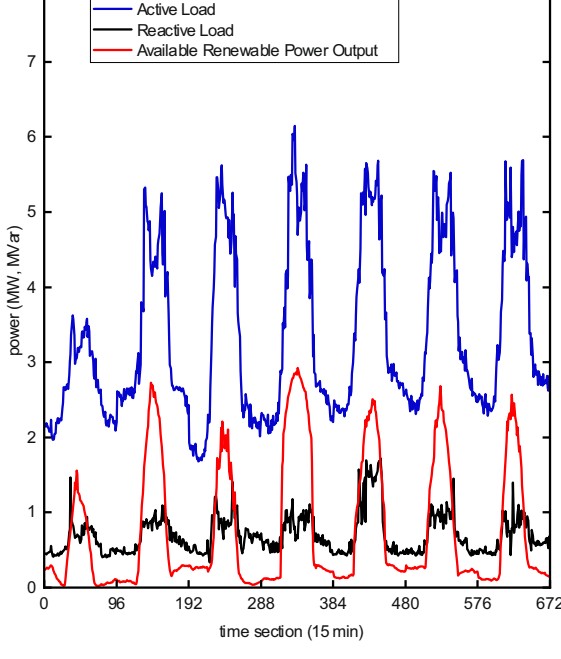

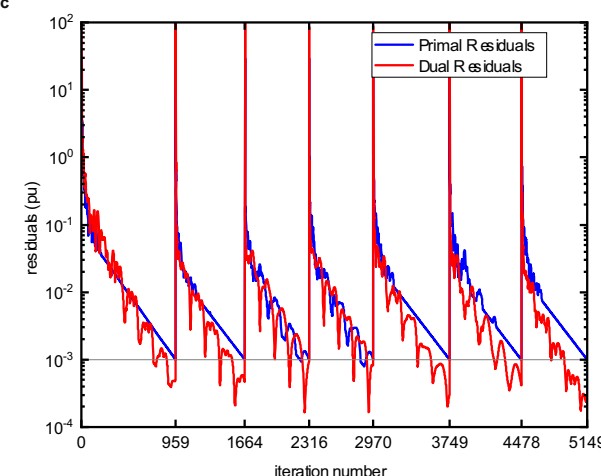

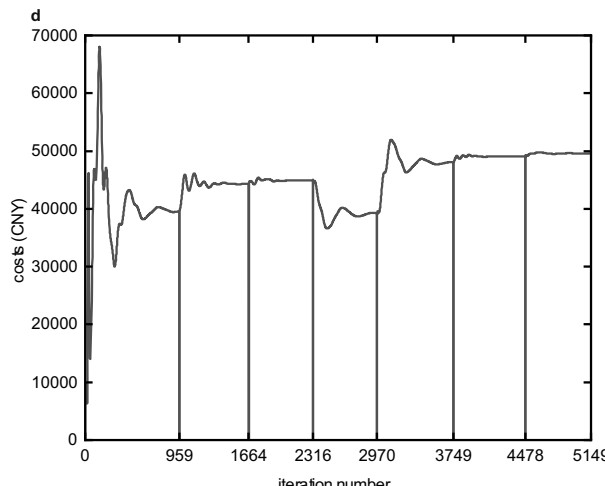

**Fig. 4 | The 60-bus distribution grid in Yingkou City, China and the iteration results. a** The general structure of the test distribution grid, which is located in the economic-technological development area of Bayuquan district, Yingkou city. **b** The active load, reactive load, and renewable power generation ceilings from 6 Nov 2023 to 12 Nov 2023. **c** The primal and dual residuals during the decentralized iteration, which respectively indicate the conflicts between neighboring DESs, i.e.,

$\max(|\mathbf{x}_{c,i,j}^{(k)} - \mathbf{e}_{c,i,j}^{(k)}|)$ and the variation of the decision in successive iterations, i.e., $\max(|\mathbf{x}_{c,i,j}^{(k)} - \mathbf{x}_{c,i,j}^{(k-1)}|)$. ($k$) denotes the iteration number. The residuals are in the per-unit system and the base value for power and voltage takes 1 MW and 1 KV. **d** The total energy cost of the decisions of all DESs during the iterations. Source data are provided as a Source Data file.

feasibility and optimality, although this also damages his own income. Such irrational participants may perform a Deny-of-Service(DoS) attack.

The attack simulations are performed on the data of 9 Nov, and the results are presented in Table 2. All of the solution schemes obtain dispatch results close to optimality when all participants are honest. It can be seen that the proposed method and state-of-the-art blockchain-only solutions are immune to manipulation behavior a and b. This is because the DES agents are only required to verifiably bid the cost functions and the final decision is made by all participants. In the centralized management case under manipulation b for comparison, we let the owner of the grid-connection point(DES

0) be the dispatcher, who tampers with the optimization results by increasing power output of generators in DES 5, 21 and 49 and setting false LMP for settlement. Although the total operation cost is only increased by 0.4%, the manipulation of LMP would benefit himself and DES 52(the distributed wind farm) with an extra income of 2221 CNY, and the other prosumers suffer a high local transaction cost due to manipulation on the LMP(see Fig. 6a). In the PA-only case under manipulation a, we let the dishonest DES owner 5 provide false decision results to its neighbors. Although in this case most DES owners' energy costs remain the same as in the honest case, and the actual final operation cost of the entire system does not increase at all, DES 6-11 are fooled with higher electricity bills, and DES owner 5's cost is decreased by 228 CNY(see Fig. 6b).

**Table 1 | Energy cost of the DESs and some statistics of the test week from 6 November to 12 November**

| Daily cost (in CNY) and statistics of the week | | | | | | | |
|---|---|---|---|---|---|---|---|
| Date | 6 Nov | 7 Nov | 8 Nov | 9 Nov | 10 Nov | 11 Nov | 12 Nov |
| Base Non-cooperative | 40916 | 46721 | 46969 | 42271 | 50658 | 51332 | 51698 |
| Proposed Decentralized | 39699 (−3.0%) | 44324 (−5.1%) | 44908 (−4.4%) | 39226 (−7.5%) | 48082 (−5.3%) | 49065 (−4.4%) | 49587 (−4.1%) |
| Theoretical Centralized[1] | 39697 | 44314 | 44893 | 39105 | 47974 | 49064 | 49584 |
| Statistics of the week | | | | | | | |
| Parameters | | | | | | Base | Proposed |
| Local generation (MWh) | | | Renewable source | | | 132.1 | 132.1 |
| | | | Gas-fired generator | | | 72.6 | 74.5 |
| Electricity purchased from the upper-level grid (MWh) | | | | | | 354.0 | 352.3 |
| Local generation cost (CNY) | | | | | | 44869 | 46537 |
| Transaction cost with the upper-level grid (CNY) | | | | | | 285696 | 246576 |

[1] We assume an ideal honest dispatcher and use the interior-point method to obtain the theoretically optimal results.

When there are irrational malicious participants, the management of DESs becomes more complicated. For the PA-only scheme and existing methods that combine PA and blockchain directly, the irrational DES participant can randomly broadcast wrong solutions to their subproblems to prevent convergence of the management process. Therefore the worst decision time may turn to infinity and the final cost is regarded as the same as in the non-cooperative case. While in the proposed or existing blockchain-only schemes, such behavior is not possible due to supervision and we only consider the DoS attack of DES owner 5 for illustration. Since DES owner 5 refuses to join the negotiation process, the final day-ahead decentralization process is executed as if DES owner 5 does not exist, and the final cost is increased by 1.5%. Still, the proposed method and state-of-the-art blockchain-only solutions can reach a suboptimal solution without the participation of DES owner 5. It should be noted that such irrational behavior will also damage the income of DES 5 by 146 CNY.

As for the time consumption, the proposed method does not have an advantage because the iterative communication and solutions to encrypted subproblems may slow down the decision process. At this cost, the security and decision robustness are guaranteed, and the participants' privacy is preserved. It is worth emphasizing that privacy protection may also hinder direct strategic bidding behavior (see Supplementary Note 5 and 6). That is, a DES with a dominant market position cannot glimpse the participants load patterns and bid parameters, and use this information superiority to simply adjust its bid cost accordingly for more profits.

### Setup of a real decentralization platform

We also set up a real communication network system(see Fig. 7) for a demonstration of the decentralized management process. Due to the lack of physical devices, we only simulate on a test 10-node system, whose information and the decentralized optimization results can be found in Supplementary Note 5. For a dispatch with 15 min resolution, as more computation parties are involved, the time consumption for each iteration reaches about 0.2 s, including 0.1 s for problem solution at the computation parties, 0.02 s for subproblem updates inside the TEE enclave and 0.08 s for communication and verification of the solutions by the miners. For a dispatch with a 1-hour resolution, the time consumption for the subproblem solution can be reduced to 0.03s due to reduction of decision variables, and the communication time is about 0.05 s.

## Discussion

In the management of multiple DESs, decentralized schemes have garnered significant attention due to their alignment with the decentralized structure of the user side. However, decentralization based solely on the blockchain can raise privacy concerns, while decentralization based on parallelizable mathematical algorithms may compromise security and decision robustness if DES agents deviate from predefined rules. This paper introduces a management mechanism for DESs that is privacy-preserving, secure, and fully decentralized by combining blockchain and parallelizable mathematical optimization methods. We leverage the strengths of these methods to address their respective weaknesses.

The concept of decentralization in this work encompasses two aspects. The first is specialization, whereby the overall management task is divided into distinct sections, each handled by dedicated professionals. The DES owners, miners, and computation parties work on specific sections. The second is distributed computation, where the calculation related to each section is decomposed and carried out by multiple participants. More importantly, all of the exchanged data within or across sections are encrypted fragments, safeguarding against the massive leaks of sensitive information and malicious analysis and manipulation.

We use an example to support the following findings. First, the proposed scheme can obtain a feasible and bid-cost-optimal operation plan through efficient iterations. Second, the proposed scheme protects the sensitive information of DESs. This may help prevent risks from other adversaries. Third, the DESs stand to benefit in the settlement stage, which encourages more participation so that the integrated system can turn more competitive.

We base the security of our framework on the integrity of edge devices to perform simple calculation and communication tasks for DESs. Although there are some widely used solutions to ensure this feature[40,41,46], in actual production environments the edge equipment may also turn offline due to network failure or fluctuations. In engineering applications, backup plans need to be set up by regulators or the owner of the distribution grid. Besides, we rely on modern asymmetric cryptography-based network infrastructures and TEE hardware producers. Their vulnerability should also be carefully considered, but is out of the scope of our research.

## Methods
### Optimization subproblems of DESs and their encryption

The overall process of the proposed scheme is shown in Fig. 2. All of the DESs are supposed to bid the cost functions and some other necessary information, such as capacities of renewable sources and maximum charge and discharge rate of batteries, to formulate the local optimization subproblem. The local optimization subproblems are deduced from the dual decomposition[47] of the integrated bid cost optimization problem of all DESs. The modeling and decomposition methods have been well discussed in previous studies and a thorough review has been made by Molzahn et al.[26]. We briefly introduce the

**a**

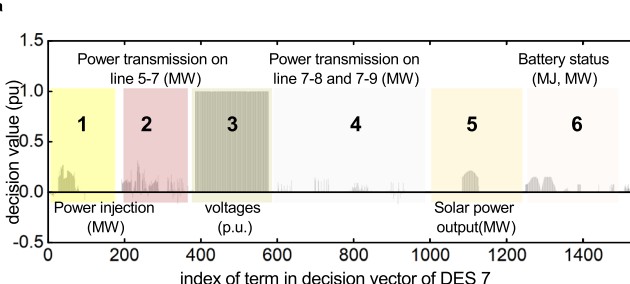

**b**

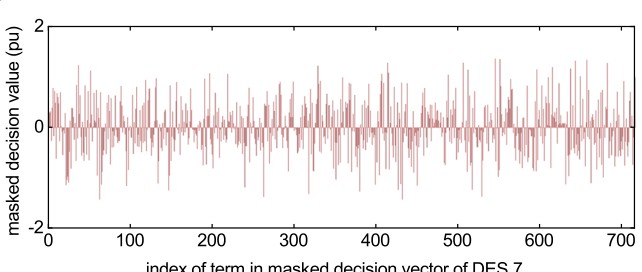

**c**

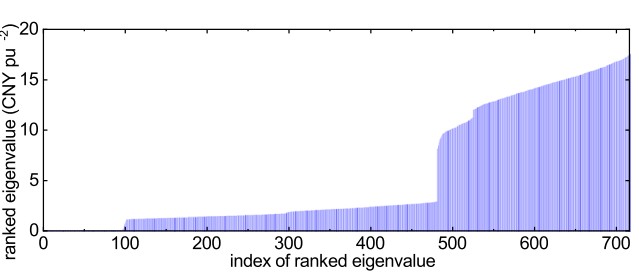

**Fig. 5 | The decision variables of DES 7 obtained on data of 6 Nov 2023. a** The optimal value of the original decision vector whose length is 1536. The original decision vector embeds important information of DES 7. **b** The optimal values of the publicly-exposed masked decision vector whose length is 716. It has a lower dimension and the values are distributed irregularly. Only DES 7 can recover the original vector. DES 5, 8, and 9 can also learn a part of the original vector. For example, the terms marked red (2) and light green (3) are available to DES 5 because they are related to power transmission between DES 7 and 5. **c** The eigenvalues of the encrypted quadratic cost function matrix $\mathbf{H}_i'$ (see Methods for definition) of DES 7 in ascending order, from which the actual bid cost functions are also hided. Source data are provided as a Source Data file.

model here, and one can still refer to Supplementary Note 1 for a more detailed explanation. Generally speaking, the subproblem for DES agent *i* appears as follows:

$$\min_{\mathbf{x}_i = [\mathbf{x}_{d,i}, \mathbf{x}_{c,i,j} \forall j \in \mathcal{N}_i]} \left( \frac{1}{2} \mathbf{x}_{d,i}^\top \mathbf{H}_{d,i} \mathbf{x}_{d,i} + \mathbf{c}_{d,i}^\top \mathbf{x}_{d,i} \right) + \tag{2a}$$

$$\sum_{j \in \mathcal{N}_i} \left[ (\boldsymbol{\lambda}_{i,j}^\top (\mathbf{x}_{c,i,j} - \mathbf{e}_{c,i,j}) + \frac{1}{2} \| \mathbf{x}_{c,i,j} - \mathbf{e}_{c,i,j} \|_{\boldsymbol{\Theta}}^2 \right] \tag{2b}$$

$$s.t. \ \mathbf{A}_{d,i} \mathbf{x}_{d,i} + \sum_{j \in \mathcal{N}_i} \mathbf{A}_{c,i,j} \mathbf{x}_{c,i,j} = \mathbf{b}_i \tag{2c}$$

$$\mathbf{G}_{d,i} \mathbf{x}_{d,i} + \sum_{j \in \mathcal{N}_i} \mathbf{G}_{c,i,j} \mathbf{x}_{c,i,j} \leq_{\mathcal{K}_i} \mathbf{h}_i \tag{2d}$$

where $\| \cdot \|_{\boldsymbol{\Theta}}^2$ stands for the quadratic form defined by diagonal matrix $\boldsymbol{\Theta}$, and inequality $\mathbf{f}(\mathbf{x}) \leq_{\mathcal{K}} \mathbf{0}$ means that $-\mathbf{f}(\mathbf{x}) \in \mathcal{K}$ where $\mathcal{K}$ is a pointed convex cone with a nonempty interior. A example of $\mathcal{K}$ is the positive orthant $\mathbb{R}_+^n$

**Table 2 | Comparisons among different operation schemes considering different malicious behaviors on 9 Nov 2023**

| | Total Energy Cost(CNY) | | | | Manipulators' gains (CNY) | | | Decision Time [1] | Privacy |
|---|---|---|---|---|---|---|---|---|---|
| | Honest | a | b | c | a | b | c | | |
| Proposed | 39226 | / | / | 39814 (+1.5%) | / | / | −146 | ~120s | Yes |
| Non-cooperative | 42271 (+7.8%) | / | / | / | / | / | / | ~1s (no communication, fast local decision) | Yes |
| Centralized[29] | 39105 (−0.3%) | / | 39379 (+0.4%) | 39814 (+1.5%) | / | 2221 | −146 | ~15s | No |
| PA only[26] | 39226 (+0.0%) | 39226 (+0.0%) | / | 42271 (+7.8%) | 228 | / | −146 | ~50s or infinity (non-convergence in case c) | Yes |
| Blockchain only[16–18] | 39105 (−0.3%) | / | / | 39814 (+1.5%) | / | / | −146 | ~40s | No |
| PA with blockchain for data aggregation[31–34] | 39226 (+0.0%) | / | / | 42271 (+7.8%) | 228 | / | −146 | ~65s or infinity (non-convergence in case c) | No |

[1] The time consumption is collected on a desktop computer with Intel i7-12700 CPU and 32GB memory by simulating the behavior of the participants and may vary on different platforms.

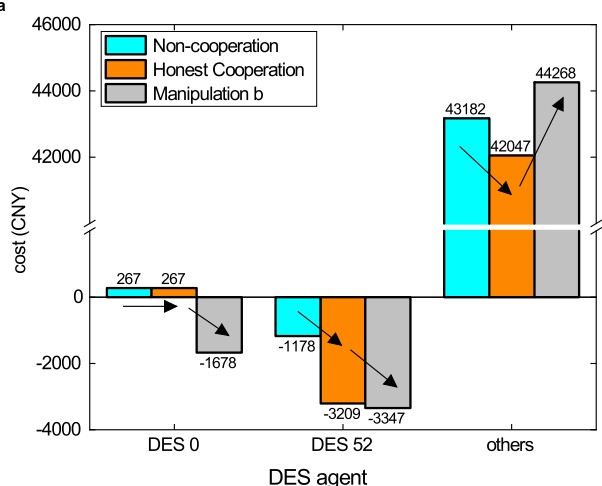
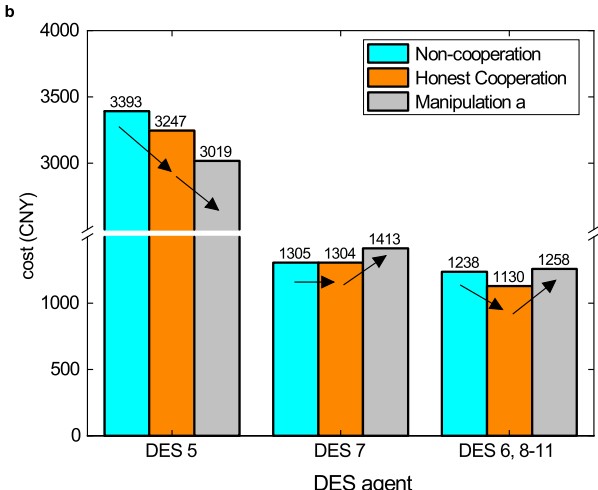

**Fig. 6 | The unfair result due to manipulation of the DES participants. a** The centralized dispatcher tampers with the optimal dispatch solution as well as the multipliers used for LMP-based settlement. This increases the transaction cost of DESs that in general serve as energy consumers by 2221 CNY(5.3%). **b** In PA, DES 5 pretends the power transmission on lines 5-6 and 5-7 would result in an extra cost and provides its neighbors with wrong but deliberately calculated subproblem solutions. Although the algorithm finally converges and other DESs are in general not influenced, the shadow prices(Lagrangian multipliers) calculated at DES 6-11 are higher and they have to pay more bills. DES 5 benefits from such manipulation behavior. Source data are provided as a Source Data file.

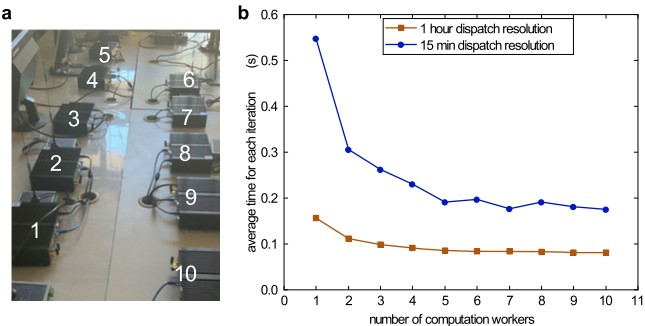

**Fig. 7 | Test results on the real communication network. a** The real communication network connecting 10 embedded computers that imitate the behavior of DES participants. 1#, 4#, 7# and 10# are selected as blockchain miners. A desktop computer with an Intel i7-12700 CPU is used to simulate the behavior of computation parties. **b** The average computation time for each iteration when the number of computation workers varies. The decentralized dispatch takes one hour resolution(24 time sections in one day) and 15 min resolution (96 time sections in one day). Source data are provided as a Source Data file.

The minimization target presented above includes two parts. The first is the bid cost of all the devices belonging to DES $i$, where $\mathbf{H}_{d,i}$ and $\mathbf{c}_{d,i}$ are parameters with proper dimensions submitted by DES $i$ and $\mathbf{x}_{d,i}$ denotes the decision variables of distributed sources, such as time-varying active power output of turbines and batteries. The second is the augmented Lagrangian term penalizing the inconformity between $i$ and its neighboring DESs $\mathcal{N}_i$. $\mathbf{x}_{c,i,j}$ encompasses the decision variables describing power transmission between DES $i$ and its neighbor $j \in \mathcal{N}_i$. For example, the power setpoint of line $ij$ and the voltage of the DES located at the parent node are included in both $\mathbf{x}_{c,i,j}$ of DES $i$ and $\mathbf{x}_{c,j,i}$ of DES $j$. $\mathbf{e}_{c,i,j}$ is the negotiated value of $\mathbf{x}_{c,i,j}$ and $\mathbf{x}_{c,j,i}$ in each iteration. Equation (2c) represents operational constraints of local devices, energy conservation, and inter-DES power flow constraints. Equation (2d) represents the security constraints of the decision variables. According to each DES's preference, personalized seller-only or timer strategies are also implemented as constraints into equation (2c)-(2d). In addition to the

cost parameters, the DESs should also bid other necessary information such as loads and generation capacities so that these constraints can be properly formulated.

The subproblem (2) can be written in a more compact form as follows:

$$\min \frac{1}{2}\mathbf{x}_i^\top \mathbf{H}_i \mathbf{x}_i + \mathbf{c}_i^\top \mathbf{x}_i + a_i \tag{3a}$$

$$s.t. \ \mathbf{A}_i \mathbf{x}_i = \mathbf{b}_i \tag{3b}$$

$$\mathbf{G}_i \mathbf{x}_i \le_{\mathcal{K}_i} \mathbf{h}_i \tag{3c}$$

$\mathbf{x}_i \in \mathbb{R}^n$ combines $\mathbf{x}_{d,i}$ and all $\mathbf{x}_{c,i,j}$. We mathematically express $\mathbf{x}_{c,i,j}$ as:

$$\mathbf{x}_{c,i,j} = \mathbf{C}_{ij}\mathbf{x}_i \tag{4}$$

The parameters $\mathbf{A}_i \in \mathbb{R}^{m \times n}$, $\mathbf{b}_i \in \mathbb{R}^m$, $\mathbf{G}_i \in \mathbb{R}^{o \times n}$, $\mathbf{h}_i \in \mathbb{R}^o$ involve private information of DES $i$, such as its load pattern, device composition and cost information. All of the parameters are invariant in the subsequent iterations except $\mathbf{c}_i$ which is related to $\boldsymbol{\lambda}_{ij}$ and $\mathbf{e}_{c,i,j}$. We enable DESs to replace $\mathbf{x}_i$ with another decision vector $\mathbf{y}_i \in \mathbb{R}^{(n-m)}$,

$$\mathbf{x}_i = \mathbf{N}_i \mathbf{R}_i \mathbf{y}_i + \mathbf{x}_i^0 \tag{5}$$

where $\mathbf{R}_i \in \mathbb{R}^{n \times n}$ is a randomly generated invertible matrix for masking. $\mathbf{N}_i \in \mathbb{R}^{n \times (n-m)}$ has full column rank and defines the null space of $\mathbf{A}_i$, i.e., $\mathbf{A}_i \mathbf{N}_i = \mathbf{0}$. $\mathbf{x}_i^0$ is a particular solution of $\mathbf{A}_i \mathbf{x}_i = \mathbf{b}_i$. These matrices and vectors are generated only once before the iterations. In respect to $\mathbf{y}_i$, we write the following optimization problem:

$$\min \frac{1}{2}\mathbf{y}_i^\top \mathbf{H}_i' \mathbf{y}_i + \mathbf{c}_i'^\top \mathbf{y}_i + a_i' \tag{6a}$$

$$s.t. \ \mathbf{G}_i' \mathbf{y}_i \le_{\mathcal{K}_i} \mathbf{h}_i' \tag{6b}$$

where,

$$\mathbf{H}_i' = \mathbf{R}_i^\top \mathbf{N}_i^\top \mathbf{H}_i \mathbf{N}_i \mathbf{R}_i \tag{7a}$$

$$\mathbf{c}_i' = \left(\mathbf{c}_i + \mathbf{x}_i^{0\top}\mathbf{H}_i\right)\mathbf{N}_i\mathbf{R}_i \tag{7b}$$

$$a_i' = \frac{1}{2}\mathbf{x}_i^{0\top}\mathbf{H}_i\mathbf{x}_i^0 + \mathbf{c}_i^\top\mathbf{x}_i^0 \tag{7c}$$

$$\mathbf{G}_i' = \mathbf{G}_i\mathbf{N}_i\mathbf{R}_i \tag{7d}$$

$$\mathbf{h}_i' = \mathbf{h}_i - \mathbf{G}_i\mathbf{x}_i^0 \tag{7e}$$

The parameters marked with apostrophes in equation (7) have masked sensitive information and can be easily calculated locally by each DES. Besides, it is easy to prove (see Supplementary Note 2) that the original cost-minimization subproblem (2) and the masked problem (6) have the same optimum, and that the optimizer of the problem (2) can be recovered by substituting that of the problem (6) into equation (5).

In each iteration, the DES owners are only required to reveal $\mathbf{H}_i'$, $\mathbf{c}_i'$, $\mathbf{G}_i'$ and $\mathbf{h}_i'$ to all of the scheme participants. It should be noted that $\mathbf{H}_i'$, $\mathbf{G}_i'$ and $\mathbf{h}_i'$ only needs to be released once while $\mathbf{c}_i'$ should be updated successively in the iterations. Besides, since $a_i'$ does not affect the solution of problem (6), it is also kept as a secret by DES $i$.

## Settlement of electricity

A settlement stage is initiated after the cooperative solution of the management problems. It should be noted that the lagrangian multiplier, or shadow prices, $\boldsymbol{\lambda}_{i,j}$ penalizes the inconsistency of decisions, i.e., $\mathbf{x}_{c,i,j} - \mathbf{e}_{c,i,j}$, where $\mathbf{e}_{c,i,j}$ can be seen as a constant after convergence. In other words, $\boldsymbol{\lambda}_{i,j}$ also directly measures the marginal cost of $\mathbf{x}_{c,i,j}$ which is related to power carried by transmission lines. Besides, from symmetry we can find that $\boldsymbol{\lambda}_{i,j} + \boldsymbol{\lambda}_{j,i} = \mathbf{0}$ always holds and $\mathbf{x}_{c,i,j} = \mathbf{x}_{c,j,i}$ after convergence. Therefore, in our design, each DES $i(j)$ should pay $\boldsymbol{\lambda}_{i,j}^\top\mathbf{x}_{c,i,j}$ ($\boldsymbol{\lambda}_{j,i}^\top\mathbf{x}_{c,j,i}$) to its neighbor $j(i)$. If a DES truthfully bid its cost functions, then its total cost can be calculated as,

$$\text{cost}_i = \left(\frac{1}{2}\mathbf{x}_{d,i}^\top\mathbf{H}_{d,i}\mathbf{x}_{d,i} + \mathbf{c}_{d,i}^\top\mathbf{x}_{d,i}\right) + \sum_{j\in\mathcal{N}_i}\boldsymbol{\lambda}_{i,j}^\top\mathbf{x}_{c,i,j} \tag{8}$$

Theoretically, an ideal settlement scheme should satisfy: (a). Individual Rationality: The participants would benefit from joining in the scheme; (b). Balanced Budget: The total payments are equal to, or at least not less than total incomes; (c). Incentive Compatibility: To tell the truth is the dominant strategy in all cases. A weaker requirement is that telling the truth is the dominant strategy when other participants also tell the truth; (d). Ex-post Pareto Efficiency: Social welfare is maximized and there is no available action that makes one individual profit more without damaging others' interests.

However, there is no such scheme according to Myerson-Satterthwaite theorem[48]. The above settlement ensures individual rationality, balanced budget, and efficiency and the proof and discussion are provided in Supplementary Note 7. Incentive compatibility does not hold, which means DES participants may report false cost functions to achieve a lower individual cost. However, the proposed scheme prevents information imparity due to privacy leakage and relieves such behavior. Also, it is possible that a DES owner checks his historical bidding information and revenues and bid strategically through a data-driven approach. This is considered normal market behavior instead of malicious manipulation because the owner does not have an unfair superior position. Discussion on such behavior is beyond the scope of this paper.

## Functions of each party in the decentralized coordination

The basic concepts of the proposed decentralization are the division of labor and mutual supervision accompanying with decomposition of the mathematical optimization model. Different individuals are designed to take different actions.

(a). (DES participants and their edge devices). Each DES owner is equipped with an edge device with TEE functionality that can be remotely attested by the blockchain. The code running in TEE cannot be tampered with by unauthorized third-parties[39]. But this promise in security also leads to performance bottlenecks[49]. Although TEEs are vulnerable to side-channel attacks, such attacks usually rely on physical contacts with the edge devices to steal private information, and may not harm the operation security. Therefore, we do not consider their threat.

The actions of DES participants and their corresponding edge devices are organized according to the algorithm of ADMM[45]. Before the iterations start, the DES owner $i$ should first provide information of local generation sources to the TEE-backed edge device. The edge device generates key matrices $\mathbf{N}_i\mathbf{R}_i$ and $\mathbf{x}_i^0$ through simple matrix operations, and informs all neighboring $j$ of the matrices $\mathbf{C}_{ij}\mathbf{N}_i\mathbf{R}_i$ and $\mathbf{C}_{ij}\mathbf{x}_i^0$. They can be used by edge device $j$ to recover the coupling decisions $\mathbf{x}_{c,i,j}$. Then in each iteration $k$, DES edge device $i$ broadcasts the parameters $\mathbf{H}_i'$, $\mathbf{c}_i'^{(k)}$, $\mathbf{G}_i'$ and $\mathbf{h}_i'$, waits for the consistent solutions $y$ from the blockchain, and calculates,

$$\mathbf{e}_{c,i,j}^{(k)} := \frac{1}{2}\left(\mathbf{C}_{ij}\mathbf{N}_i\mathbf{R}_i\mathbf{y}_i^{(k)} + \mathbf{C}_{ij}\mathbf{x}_i^0 - \mathbf{C}_{ji}\mathbf{N}_j\mathbf{R}_j\mathbf{y}_j^{(k)} - \mathbf{C}_{ji}\mathbf{x}_j^0\right) \tag{9}$$

$$\boldsymbol{\lambda}_{i,j}^{(k)} := \boldsymbol{\lambda}_{i,j}^{(k-1)} + \boldsymbol{\Theta}(\mathbf{x}_{c,i,j}^{(k)} - \mathbf{e}_{c,i,j}^{(k)}) \tag{10}$$

Afterward, the parameters of the masked problem (6) can be updated according to equation (2), equation (3), and equation (7), and iteration $k+1$ begins. The detailed algorithm is presented in Supplementary Note 8.

(b). (Blockchain Miners). The blockchain miners collect subproblems submitted by DESs, re-allocate them to workers, and verify the solutions given by workers. In the test case, the blockchain miners are organized through the pBFT[15] protocol, which tolerates arbitrary Byzantine behavior of no more than 1/3 miners. The miners communicate with each other and follow a multi-phase protocol to reach a consensus on an ordered series of data. In pBFT, one agent serves as the leader to initiate data operations and the others are replicas. Only when the operation has been 'prepared' and 'committed' by enough replicas can it be executed. If the leader is suspected of cheating, crashing, or network failure, the replicas follow a 'view-change' protocol to elect a new leader. The leading and the following miners' blockchain maintenance modules can be found in Supplementary Note 9.

(c). (Computation Parties). The computation parties receive problems distributed by miners and solve them with an interior-point method implemented by themselves or existing open-source or commercial solvers.

## Other implementation details

On the test system in Yingkou City, we simulate the solution process on a 32GB Intel i7-12700 computer. We use the SCONE[50] community version's API to set up SGX enclaves in simulation mode, and use the numpy[51] module for the formulation of the subproblems and the calculations. CVXPY[52] and GUROBI[53] solver are employed to solve the optimization subproblems. The decision variables are all initialized to 0. The Lagrangian parameters for active power transmission is initialized to TOU prices and the Lagrangian parameters for other coupling constraints are all initialized to 0.

On the established real communication network, the WebSocket protocol[54] is adopted for communication. The participants' web

applications are implemented in the Python Flask[55] backend framework. The same desktop computer is used for the solution of the subproblems.

## Reporting summary

Further information on research design is available in the Nature Portfolio Reporting Summary linked to this article.

## Data availability

The data used in this study are deposited https://github.com/sqinghan/PAchain.git here, and are also available from the authors upon request. The data generated in this study are provided in the Source Data file. Source data are provided with this paper.

## Code availability

The code used in this study is available https://github.com/sqinghan/PAchain.git here, and is also available from the authors upon request.

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

## Acknowledgements

This work is supported by the National Natural Science Foundation of China under Grant 52125604 (Prof. Qun Chen). We are grateful to Prof. Zheng Yan from Xidian University for providing valuable advice. We would like to thank State Grid Liaoning Electric Power Supply Co., Ltd. and State Grid Liaoning Electric Power Research Institute for providing data on the test system.

## Author contributions

Q.S.: conceptualization, methodology, instantiation, data acquisition, and result analysis; H.M.: methodology, instantiation, data acquisition, and result analysis; T.Z and Y.X.: instantiation, data acquisition and result analysis; Q.C. conceptualization, methodology, and supervision. All authors contributed to writing and refining the manuscript.

## Competing interests

The authors declare no competing interests.
