## [Peer Review File · Nature Communications]

Break down the Decentralization-Security-Privacy Trilemma in Management of Distributed Energy SystemsEditorial Note: Figure 1 on page 12 in this Peer Review File has been amended to remove third-party material where no permission to publish could be obtained.

REVIEWER COMMENTS

Reviewer #1 (Remarks to the Author):

The paper proposes a solution to address Decentralization, Security and Privacy by combining blockchain, distributed optimization, and encryption. While the combined use of all three techniques may be new, combined use of blockchain and distributed optimization methods has been used in several papers in the past three years, just in the field of power trading or energy trading.

- 1, Please enhance literature review, and state your contribution against those that combined blockchain and distributed optimization.
- 2, The case study are advised to be run using a realistic system.
- 3, The case study presents some figures that show the results of the proposed method, but lacks numerical comparison against those in Fig2 to highlight its merit. For example, the merit of using blockchain in defending malicious behaviour? the merit of using distribution optimization and encryption in protecting privacy?

Reviewer #2 (Remarks to the Author):

This article presents a nine-step method for a distributed energy system (DES) optimization framework using blockchain technology. The method begins with DES owners providing sensitive information about their devices (Step 1), followed by the formulation of encrypted optimization subproblems (Steps 2-3). These subproblems are then broadcasted to the blockchain network (Step 4), in which certain computing nodes solve them (Steps 5-6). The results are verified and recorded in the blockchain by miners (Step 7). Any inconsistencies in the solutions are identified and addressed in an iterative process (Step 8), culminating in a settlement stage where final transactions are executed (Step 9). This approach aims to ensure secure, private, and efficient energy transaction management in a decentralized environment. However, I have identified certain aspects that might require significant attention.

1. Iterative Computation Process:

The iterative process completes when a majority of miners agree on convergence, indicating the elimination of decision inconsistencies (pages 4-6 in the paper). However, this will impose a vulnerability when there are dishonest DES participants. If any DES consistently provides inaccurate or false data, the system could be trapped in an endless loop of iterations without reaching a consensus on accurate data. This issue will be exacerbated by time constraints in energy market transactions, where dishonest participants might deliberately prolong computation beyond transaction deadlines, rendering the process ineffective and without any repercussions for their dishonesty.

2. Non-Performing Computing Nodes:

The framework relies on distributed computing nodes to solve encrypted DES optimization subproblems. In a realistic blockchain network, not all the computing nodes might act honestly or diligently perform their tasks. Failure to execute computation tasks by some nodes, either due to dishonest intentions or

operational inefficiencies, could result in incomplete resolution of optimization subproblems, preventing the system from obtaining a complete and accurate solution.

3. Byzantine Leader Node in PBFT:

In the PBFT consensus protocol, each consensus round requires the selection of a leader node. If a Byzantine node is chosen as the leader, it could manipulate the assignment of computing tasks, preferentially allocating subproblems to dishonest computing nodes. These nodes, in collusion with the Byzantine leader, might deliberately fail to compute their assigned encrypted subproblems, leading to delays or inaccuracies in the optimization results. The subtlety of this manipulation makes it challenging for honest miners to detect the Byzantine nature of the leader or the intentional assignment of tasks to dishonest nodes.

Reviewer #3 (Remarks to the Author):

This paper targets the challenges of security and privacy for distributed energy systems management. It devises a blockchain-based approach whereby participants obfuscate their energy information and jointly manage the state of the DES without revealing their private data.

This problem is timely and highly relevant with the increase in DES penetration towards emissions reduction targets. The reviewer agrees with the high level approach for solving this problem in a decentralised way. However, there are several points that need improvement or are unclear in the manuscript, as detailed below.

The trilemma is mentioned in the title, and is implicitly mentioned there. In the introduction, it is used without explicit definition. There needs to be much deeper discussions of the trilemma and the challenges involved in satisfying all three requirements.

The abstract mentions a 5.45% cost reduction. What is the significance of this reduction in the context of the target systems?

The manuscript writing needs significant improvement. For instance, the first paragraph of the introduction refers to 'mortals', which is surprising. The last sentence of the same paragraph needs to be reworded. There are several references in sentences that are ambiguous such as on line 44 (this end), line 83 (unauthorized ones), line 101 (through subtle design - unclear what this refers to), and many others.

The manuscript needs to be clear upfront about the timescales it is targeting. It becomes evident later in the manuscript that it aims for day-ahead DES management, but this should be brought forward, to avoid perceptions that real-time management may be possible with this approach.

References to trusted computing base are not well defined.

Overall, the summary of the approach from lines 97 to 115 is not clear and leaves the reviewers unsure of what the approach actually does.

Line 145: it not clear whether the edge equipment jointly or individually formulate the subproblems. This is a very important point.

Line 148: augmented Lagrangian based penalties are not defined

Step 3 in Figure 1: the partial keys exchanged by neighbours need to be better defined.

Step 7 in Figure 1: it is not clear who runs the solutions whose optimality is checked

Line 164-166: the problem inconsistencies, the coupled results are not clear

Author Response of

Break down the Trilemma in Management of Distributed Energy Systems: Decentralization, Security and Privacy

Qinghan Sun, Huan Ma, Tian Zhao, Yonglin Xin, Qun Chen*
Nature Communications

RC: Reviewer Comment, AR: Author Response, Manuscript text

Dear reviewers,

Thank you for the valuable comments and suggestions concerning our manuscript [NCOMMS-23-37463A-Z] "Break down the Trilemma in Management of Distributed Energy Systems: Decentralization, Security and Privacy". Our manuscript has been thoroughly revised:

- The literature review is supplemented with existing studies that have combined distributed algorithms and blockchain.
- The management framework is slightly adjusted and the explanations in section **Formulate, encrypt and distribute privacy-preserving optimization tasks to workers through blockchain** are rephrased.
- The proposed method is tested on a real 60-bus distribution grid in Yingkou City, China. This system is much bigger than the previous one and we study the performance of the proposed method on the data from 6 Nov 2023 to 12 Nov 2023.
- We compare the proposed method with other management schemes and under several manipulative behaviours and show how the proposed method ensures a secure and privacy-preserving decentralized coordination.

Please find our itemized responses in below and the revised manuscript in the re-submitted files. We sincerely hope that the responses and the revised manuscript have addressed all your comments and suggestions.

Authors' responses to Reviewer 1

The general comments from the reviewer 1:

The paper proposes a solution to address Decentralization, Security and Privacy by combining blockchain, distributed optimization, and encryption. While the combined use of all three techniques may be new, combined use of blockchain and distributed optimization methods has been used in several papers in the past three years, just in the field of power trading or energy trading.

RC: *1. Please enhance literature review, and state your contribution against those that combined blockchain and distributed optimization.*

AR: We gratefully appreciate for your valuable suggestion. We have supplement the literature review in the **Main** section. Herein, we briefly state the contributions and differences.

So far, many researchers have realized the security risks of distributed optimization algorithms and try to use blockchain to mend this gap. However, they mainly use the blockchain to ensure consistency. That is, the dishonest participant cannot fool other participants with different information to deceive them to make suboptimal decisions. Depending on the specific algorithms that are used, aggregated data of local primal decisions(the operation points of generation devices) or dual decisions(the price signals) may be made public in a tamper-proof manner. Two problems arise with such simple combination of distributed algorithms and blockchain:

1. Even when the consistency of information provided by distributed energy system(DES) agents is ensured by blockchain, the DESs are allowed to locally formulate and solve the subproblems and this leave a loophole. For instance, a dishonest DES can formulate false local optimization subproblems that involve false costs functions about inter-DES transmission line power. Solving the false subproblems may lead to a false higher final cost and locational marginal price, and the dishonest DES benefits from this(see the simulation of attacks in the revised manuscript). Besides, irrational malicious participants can still prevent the convergence of the iterations.
2. The topology of the power grid, the DES's decisions sensitivity to the price signals, and sometimes the plaintext of decision variables, may be disclosed on the blockchain. This cause privacy problems. From the private data, powerful energy producers may construct an information superiority and abuse its dominant market position to earn more profits.

Therefore, to ensure security, the DES participants should be only data providers instead of decision-makers without supervision. The formulation and solution of the local optimization subproblems, as well as other calculations in the iterations should be protected, but the privacy information should also not be compromised. Besides, considering the accessibility of decentralized management to lightweight energy users, complex computations of optimization problems that usually relies on commercial solvers for fast solution should also be reduced for the terminal prosumers.

In our work, we use edge devices with Trusted Execution Environment(TEE) attested by the blockchain to formulate the subproblems. They will be encrypted and offloaded to third-party computation workers. Therefore, DES participants are no longer able to cheat on the optimization problems or worry about efficient solutions to them. The blockchain attests the integrity of the edge devices and optimality of subproblem solutions, which guarantees robust convergence of the iteration. As a result, the final optimization results and the privacy are both protected.

The TEE technology used here is just an example for integrity protection of general purpose calculation, which has commercial supports from Intel or ARM. We can also design and deploy special embedded smart meters with simple matrix multiplication functionalities and hardened physical shells for practical use.

RC: 2. *The case study are advised to be run using a realistic system.*

AR: We gratefully appreciate for your valuable suggestion. We have collected data of a real 10-kV distribution grid in Bayuquan District, Yingkou City, Liaoning Province, China to perform the case study. The distribution grid is located at the economic development area of Yingkou City and we model it as a 60-bus(60 independent DESs) grid with industrial and commercial prosumers. We test the proposed decentralized management framework on the data from 6 Nov 2023 to 12 Nov 2023. Besides, the dispatch interval for decentralized management is decreased from 1 hour to 15min and the number of time sections in one day is increased to 96.

The results show that the proposed method performs robustly to obtain optimal management solutions in the test week. Please refer to the revised manuscript for further details. The **Instantiation of decentralization: test case, computation performances and economic benefits** section has been completely re-written.

RC: *3. The case study presents some figures that show the results of the proposed method, but lacks numerical comparison against those in Fig2 to highlight its merit. For example, the merit of using blockchain in defending malicious behaviour? the merit of using distribution optimization and encryption in protecting privacy?*

AR: Thank you for the comments and we have made the following modifications.

In **Analysis of privacy protection and adversarial behaviours** section of the revised manuscript, we have modelled three typical scenarios, to show how popular distributed algorithms or blockchain-only schemes are vulnerable to participants malicious attacks. Table. 4 and Fig. 7 in the revised manuscript have shown comparisons with existing decentralized management schemes. Although on the test system some manipulative behavior will not lead to a great total cost increase, the dishonest manipulators can earn themselves more profits while damaging the interest of other participants. The comparisons can show the merit of using the proposed method to defend from malicious behaviour.

- a. (dishonest DES agent) The DES owners may be required to perform some local calculations and communicate with their neighbours. The dishonest DES owners may provide wrong or inconsistent calculation results for more profits. They are rational, and thus will not try to ruin the decentralized management process.
- b. (manipulative third-party decision-maker) Some third-party participants other than DES owners, may be required to optimize the operation of the DESs. However, the manipulative decision-maker colludes with some DES and tries to make tendentious operation plans or cheat on the energy prices.
- c. (irrational malicious participant) The irrational participant just want to prevent the management process from reaching feasibility and optimality, although this also damages his own income. Such irrational participants may perform Deny-of-Service(DoS) attack.

As for privacy, the distributed optimization and encryption avoids leakage of plaintext sensitive information. This prevents a very simple manipulation of the market based on information superiority. For example, dishonest miners may disclose other DES owners' generation and load information to the dishonest DES participant, who can bid the DES parameters accordingly and strategically to acquire advantage in dispatch and settlement. This can happen even if the final dispatch decision is made by the blockchain. In the original manuscript we present this in a separate section and now it is moved to the revised supplementary materials. The discussion actually explains why we need to protect privacy.

Authors' responses to Reviewer 2

The general comments from the reviewer 2:

This article presents a nine-step method for a distributed energy system (DES) optimization framework using blockchain technology. The method begins with DES owners providing sensitive information about their devices (Step 1), followed by the formulation of encrypted optimization subproblems (Steps 2-3). These

subproblems are then broadcasted to the blockchain network (Step 4), in which certain computing nodes solve them (Steps 5-6). The results are verified and recorded in the blockchain by miners (Step 7). Any inconsistencies in the solutions are identified and addressed in an iterative process (Step 8), culminating in a settlement stage where final transactions are executed (Step 9). This approach aims to ensure secure, private, and efficient energy transaction management in a decentralized environment. However, I have identified certain aspects that might require significant attention.

RC: *1. Iterative Computation Process: The iterative process completes when a majority of miners agree on convergence, indicating the elimination of decision inconsistencies (pages 4-6 in the paper). However, this will impose a vulnerability when there are dishonest DES participants. If any DES consistently provides inaccurate or false data, the system could be trapped in an endless loop of iterations without reaching a consensus on accurate data. This issue will be exacerbated by time constraints in energy market transactions, where dishonest participants might deliberately prolong computation beyond transaction deadlines, rendering the process ineffective and without any repercussions for their dishonesty.*

AR: Thank you for your question. We agree with the reviewer that leaving too much computation authority to the DES owners may impose a vulnerability. This is a key problem in existing decentralized management schemes that only rely on distributed parallelizable algorithms(PA) or simply combine PA and blockchain, where DES participants are required to continuously solve subproblems.

However, in the proposed scheme, DES owners are only data providers and they only bid the device parameters once. Special edge devices are introduced to check the input validity, encrypt the data, and perform calculations required in the iterative computation process. Provided the input validity from the DES owners and integrity of the edge devices, the convergence can be guaranteed. In the **Analysis of privacy protection and adversarial behaviours** section of the revised manuscript, we show that dishonest participants can lead to non-convergence in existing PA-based approaches, while the proposed method avoids this by limiting the behavior of DES participants.

The reviewer may then be suspicious about the reliability of the edge devices. With our design, the calculations in the edge devices would only involve finite fundamental matrix addition and multiplications. Therefore, cheap embedded single-board computers or smart meters with hardened physical shells can be effective enough to prevent the dishonest DES participants. In the revised manuscript, we introduce Trusted Execution Environment(TEE) for deployment of edge devices. TEE is a segregated area of CPU and memory to ensure integrity and privacy of the code and data. So far, mature commercial TEE solutions include Intel SGX and ARM TrustZone. With modern TEE's support, the system would also not be trapped in endless loops. The TEE-backed edge devices can be remotely attested by blockchain miners to ensure integrity.

It should be noted that although existing commercial TEE is designed for general computation, the memory usage is usually limited(128MB as claimed by Intel SGX) and computation burden may be greatly increased due to encryption of the memory. Therefore, directly letting the DES owners to use TEE-backed computers to securely solve CPU and memory-intensive optimization subproblems would be impractical for the lightweight users and that is why we introduce third-party computation workers and miners to solve and verify the subproblems.

The reviewer also mentions that dishonest participants may deliberately prolong computation beyond deadlines. In the revised manuscript, we let the grid owner be backup of the DES owners. In Step 1, alternative subproblems will also be generated by the grid owner. If the dishonest participant provides nothing about its load and generation capacity in Step 1, or disconnect the Internet to prolong computation and communication in the iterations for too many times, the alternative subproblem will be handled by the blockchain and computation workers instead. In this case, the iterations will converge to (sub)optimality as if the dishonest

DES does not exist.

In **Analysis of privacy protection and adversarial behaviours** section of the revised manuscript, we show that either trying to ruin the convergence or keeping silent will damage the manipulator's income. Therefore, if we assume that DES owners are rational, they will not perform these dishonest behaviours. Even if they do so, the consensus process to (sub)optimality cannot be interfered. The comparisons in **Analysis of privacy protection and adversarial behaviours** show that the proposed scheme prevent well-designed manipulative actions for unjust enrichment, and also adapts to irrational dishonest behaviours.

RC: *2. Non-Performing Computing Nodes: The framework relies on distributed computing nodes to solve encrypted DES optimization subproblems. In a realistic blockchain network, not all the computing nodes might act honestly or diligently perform their tasks. Failure to execute computation tasks by some nodes, either due to dishonest intentions or operational inefficiencies, could result in incomplete resolution of optimization subproblems, preventing the system from obtaining a complete and accurate solution.*

AR: In the proposed scheme, the blockchain leader maintains a timer τ_{ite} to ensure liveness of the iterations. If some computation parties are not able to provide solution results in time limit t_{ite} , still a new block will be organized by the blockchain leader using the subproblem solutions in the last iterations, and the time limit t_{ite} will be multiplied by two. As long as at least one honest computation party is available, the iterations can converge to optimality. However, as the reviewer has mentioned, not all computing nodes act honestly or diligently, and this may slow down the solution process when the computation tasks are just equally allocated to the workers at random.

In the revised manuscript, we slightly adjust the task allocation part by introducing rating and competition. The miners are required to maintain a success rate list on the blockchain. The list will be set up as a reference for task allocation. If a computation party fails to solve the allocated subproblem tasks, its success rate will decrease and it will be allocated fewer computation tasks later. Besides, the computation workers with top ranks are required to serve as redundant backups for solutions to subproblems. In addition to the basic tasks, the top-ranked computation parties are also invited solve extra subproblems. This is not compulsory, but these computation parties are rewarded to do so.

In the revised manuscript we just present a simple example to encourage the computing nodes. More reward and punishment mechanisms can be introduced in the task allocation and solution procedure. For example, trusted timestamping can be introduced to assess the solution time of the computation parties and the time can also be designed as a reference for task allocation. It should be noted that the computation nodes cannot interfere with the fact of final convergence and the dispatch results. They can only try to slow down the process. However, their performance can be quantitatively assessed and they are also fungible. Their negative behaviour, either due to dishonest intentions or insufficient capacities, would be ruled out by a well-organized reward and punishment system and only have a limited impact on solution speed.

RC: *3. Byzantine Leader Node in PBFT: In the PBFT consensus protocol, each consensus round requires the selection of a leader node. If a Byzantine node is chosen as the leader, it could manipulate the assignment of computing tasks, preferentially allocating subproblems to dishonest computing nodes. These nodes, in collusion with the Byzantine leader, might deliberately fail to compute their assigned encrypted subproblems, leading to delays or inaccuracies in the optimization results. The subtlety of this manipulation makes it challenging for honest miners to detect the Byzantine nature of the leader or the intentional assignment of tasks to dishonest nodes.*

AR: In the original manuscript, The PBFT leader node do not allocate tasks to computation parties at will, but randomly with the hash value of previous block as the random seed. Therefore, it is not possible that a

Byzantine leader deliberately allocate tasks to dishonest computing nodes. In the revised manuscript, we further include rating and competition, and maintain a success rate table as a reference for task allocation. Computing nodes with higher capability and success rate will be allocated more tasks.

There is also a possibility that the dishonest leader may deliberately ignore the computation results from honest workers to lower down their success rates. In this case, since the computation results are broadcast in the network, the honest followers may designate the leader as having network errors and overthrow the dishonest leader.

Authors' responses to Reviewer 3

The general comments from the reviewer 3:

This paper targets the challenges of security and privacy for distributed energy systems management. It devises a blockchain-based approach whereby participants obfuscate their energy information and jointly manage the state of the DES without revealing their private data.

This problem is timely and highly relevant with the increase in DES penetration towards emissions reduction targets. The reviewer agrees with the high level approach for solving this problem in a decentralised way. However, there are several points that are need improvement or are unclear in the manuscript, as detailed below.

RC: *1. The trilemma is mentioned in the title, and is implicitly mentioned there. In the introduction, it is used without explicit definition. There needs to be much deeper discussions of the trilemma and the challenges involved in satisfying all three requirements.*

AR: Thanks for the suggestion. In the revised manuscript, we explain in detail the trilemma among decentralized management, privacy and decision security.

The above technologies focus on different concerns of decentralization and thus indicate respective limitations. Briefly speaking, privacy and secured decision are two somewhat contradictory goals in the context of decentralization, as illustrated in Fig. 1. The path toward privacy in decentralized management emphasizes on high-degree distributed computing. Such reliance on localization might be more susceptible to modifications from the DES owners. The other path toward decision and trading security asks for public confirmation and voting, which leads to privacy disclosure. So far, several studies have realized the security risks in PA and introduced blockchain to enhance integrity of the management process[31,32,33,34,35]. They use blockchain to aggregate the decision results provided by DES participants according to different mathematical algorithms and endorse its consistency, i.e., a DES participant cannot fool other participants with different results. Such data aggregation in plaintext compromises the privacy-preserving feature of PA. In [36], price signals are designed as the only information exchanged to protect privacy. However, the method lacks the support to DESs with diverse energy supply characteristics, and the power transmission process and related physical constraints are not considered. Apart from these concerns, dishonest individual participants may still deliberately disrupt the convergence process to hazard security. To solve the trilemma among decentralized coordination, security and privacy, not only the data transferred on the public blockchain should be obfuscated and verifiable, but also the off-chain calculations should be tamper-proof.

Figure 1: Trilemma in management of DESs: decentralization, privacy, and security.

RC: *The abstract mentions a 5.45% cost reduction. What is the significance of this reduction in the context of the target systems?*

AR: Thank you for the question. We mistook the numbers in the original draft and we are sorry for this negligence. Herein, 5.45% is the cost reduction of one single DES when it can glimpse privacy of other participants and bid targeted parameters. The actual cost reduction of the original test system of the proposed method compared with the non-cooperative case is about 26%(about 15000 CNY lower cost). This is a considerable amount for the test system. Such reduction indicates two points:

1. The DESs are operated with higher energy efficiencies, and the generation costs are reduced. If one DES is equipped with plenty of renewable sources and there is a risk of curtailment, it can negotiate with the other DESs to use their batteries or heat pumps for renewable accommodation. If some owners are equipped with complicated multi-energy systems, they can also negotiate with each other on the power output values and reach a higher global efficiency.
2. The transaction costs are reduced. With the proposed method, the separated DESs are united together to deal with the upper-level grid or market. This also offers a possibility for efficient demand-side response.

The cost reduction would vary according to specific conditions of the connected system. The original test system is an ideal one with high penetration of distributed energy systems. There are many optional multi-energy sources and storage equipments and the connected 10 DESs can operate in self-sufficiency. On such a distribution grid, the cost reduction of 26% or even more is possible. However, if the terminal users and DES participants lack flexible energy sources or batteries, the cost reduction would also be low.

In the revised manuscript, we use the data of a 10kV distribution grid in Yingkou City, China and the proposed method is tested from 6 Nov 2023 to 12 Nov 2023. Due to lack of heat data, we did not include heat generation and loads in the new test system but only consider the power sources and batteries. The test results show that

the expected daily cost reduction would vary from 3.0-7.5%, and the total cost reduction during the week would be 18674 CNY(-5.64%).

RC: *The manuscript writing needs significant improvement. For instance, the first paragraph of the introduction refers to 'mortals', which is surprising. The last sentence of the same paragraph needs to be reworded. There are several references in sentences that are ambiguous such as on line 44 (this end), line 83 (unauthorized ones), line 101 (through subtle design - unclear what this refers to), and many others.*

AR: Thank you for the suggestion. We have carefully revised the manuscript and correct some typos and inappropriate or inaccurate expressions. Please refer to the revised manuscript for more details.

RC: *The manuscript needs to be clear upfront about the timescales it is targeting. It becomes evident later in the manuscript that it aims for day-ahead DES management, but this should be brought forward, to avoid perceptions that real-time management may be possible with this approach.*

AR: This manuscript mainly focuses on day-ahead management. In the revised manuscript, we mention this at the start of the **Results** section.

The average time consumption for the proposed method to reach a consensus in day-ahead management(15 min interval, 96 decision points per day) of the test system is about 2 min. Considering possible manipulative behaviours, possible low convergence efficiencies and various potential emergencies, we expect the total time consumption in the worst case to be less than 15min. In the real-time management, operating decisions have to be made more efficiently. There are two possible approaches, 1. adapt the proposed framework to a faster one applicable to real-time management, 2. propose a new framework compatible to the existing day-ahead management scheme. At present, we are still working on it and this will be a focus of our further research.

RC: *References to trusted computing base are not well defined.*

AR: Thank you for the suggestion, We have revised the reference source and carefully check all of the references.

RC: *Overall, the summary of the approach from lines 97 to 115 is not clear and leaves the reviewers unsure of what the approach actually does.*

AR: Thank you for the comments and we have revised this summary

In this study, we propose a new mechanism to address the privacy and security concerns in decentralized management of DESs. The basic philosophy follows state-of-the-art decentralization solutions, where optimization subproblems are formulated for DES participants and iteratively solved to reach a consensus. But the key lies in the actual transferred data model and on-chain and off-chain calculation designs. To protect the DESs' cost functions and equipment capacities, the subproblems are encrypted before being sent to the blockchain, which can then be efficiently solved off-chain by competent computation parties and verified on-chain by miners. To avoid adversarial behaviour of DES owners, we reduce their local computation to fundamental arithmetic operations of matrices so that hardware-backed edge devices using Trusted Execution Environment(TEE)[37] are deployed to avoid disobedience to protocols. On the data of a real 10kV distribution grid in Yingkou city, China, we compare the proposed method with other decentralization approaches considering potential manipulative actions. We also set up a real communication network to demonstrate the feasibility and effectiveness of the framework.

RC: *Line 145: it not clear whether the edge equipment jointly or individually formulate the subproblems. This is a very important point.*

AR: Thank you for the question. We first present the subproblems for a better explanation. The original subproblems have the following form,

$$\min_{x_i=[x_{d,i}, x_{c,i,j}, \forall j \in \mathbb{N}_i]} \left(\frac{1}{2} x_{d,i}^\top H_{d,i} x_{d,i} + c_{d,i}^\top x_{d,i} \right) + \sum_{j \in \mathbb{N}_i} \left[(\lambda_{i,j}^\top (x_{c,i,j} - e_{c,i,j}) + \frac{1}{2} \|x_{c,i,j} - e_{c,i,j}\|_{\Theta}^2) \right] \quad (1a)$$

$$s.t. \quad A_{d,i} x_{d,i} + \sum_{j \in \mathbb{N}_i} A_{c,i,j} x_{c,i,j} = b_i \quad (1b)$$

$$G_{d,i} x_{d,i} + \sum_{j \in \mathbb{N}_i} G_{c,i,j} x_{c,i,j} \leq_{K_i} h_i \quad (1c)$$

where $x_{d,i}$ is the decision variables concerning local devices and $x_{c,i,j}$ is the decision variables concerning power transmission between DES i and j . The original problem is compacted to the following one,

$$\min \quad \frac{1}{2} x_i^\top H_i x_i + c_i^\top x_i + a_i \quad (2a)$$

$$s.t. \quad A_i x_i = b_i \quad (2b)$$

$$G_i x_i \leq_{K_i} h_i \quad (2c)$$

whose encryption have the following form,

$$\min \quad \frac{1}{2} y_i^\top H'_i y_i + c'_i{}^\top y_i + a'_i \quad (3a)$$

$$s.t. \quad G'_i y_i \leq_{K_i} h'_i \quad (3b)$$

where

$$H'_i = R_i^\top N_i^\top H_i N_i R_i \quad (4a)$$

$$c'_i = \left(c_i + x_i^{0\top} H_i \right) N_i R_i \quad (4b)$$

$$a'_i = \frac{1}{2} x_i^{0\top} H_i x_i^0 + c_i^\top x_i^0 \quad (4c)$$

$$G'_i = G_i N_i R_i \quad (4d)$$

$$h'_i = h_i - G_i x_i^0 \quad (4e)$$

N_i , R_i and x_i^0 are the key matrices that are used to encrypt or recover the actual decision variables.

In the compacted form (2), H_i , c_i , a_i , A_i , b_i , G_i and h_i are vector or matrix parameters that determine the subproblem. The edge equipment INDIVIDUALLY formulate H_i , a_i , A_i , b_i , G_i and h_i , as well as the keys (N_i , R_i and x_i^0) and the encryptions (H'_i , a'_i , G'_i and h'_i). These terms no longer change in the iterations.

As for the linear term c_i and c'_i , its formulation is a bit different and it should be updated in the iterations by the edge devices. Specifically, in the preparative steps the neighbouring edge devices EXCHANGE several rows of the encryption matrices as partial keys. Such information exchange is executed ONLY ONCE and non-iteratively. With the partial keys, c'_i can be updated in later iterations INDIVIDUALLY by each edge device according to the data from blockchain and no further inter-DES communications are needed.

RC: *Line 148: augmented Lagrangian based penalties are not defined*

AR: In the revised manuscript, we have moved forward the definition from **Methods** section to **Formulate, encrypt and distribute privacy-preserving optimization tasks to workers through blockchain** section. The $\sum_{j \in \mathbb{N}_i} [\lambda_{i,j}^T (x_{c,i,j} - e_{c,i,j}) + \frac{1}{2} \|x_{c,i,j} - e_{c,i,j}\|_{\Theta}^2]$ term in (1) is the augmented Lagrangian-based penalties.

RC: *Step 3 in Figure 1: the partial keys exchanged by neighbours need to be better defined.*

AR: The partial keys refers to some rows of the matrix keys $N_i R_i$ and x_i^0 . Specifically, if we represent $x_{c,i,j}$ as $C_{i,j} x_i$ where $C_{i,j}$ is composed of 0 and 1 and extracts rows of x_i , then the keys sent to j is $N_i R_i C_{i,j}$ and $N_i R_i x_i^0$. These keys are exchanged only once in the entire process.

RC: *Step 7 in Figure 1: it is not clear who runs the solutions whose optimality is checked*

AR: The computation parties solve the problems to obtain primal solutions(the decision variables) and the dual solutions(the Lagrangian multipliers). The blockchain miners check the optimality of these solutions.

In the revised manuscript, we have modified Figure 1 and clarify some ambiguities.

RC: *Line 164-166: the problem inconsistencies, the coupled results are not clear*

AR: Thank you for the question. We have moved forward their detailed definition from the **Methods** section to **Formulate, encrypt and distribute privacy-preserving optimization tasks to workers through blockchain** section. To be specific, $x_{c,i,j} - e_{c,i,j}$ in Eq. (1) is the problem inconsistencies, where $x_{c,i,j}$ is the coupled decision between DES i and j (e.g., the power transmission on line ij), and $e_{c,i,j}$ is the average value of $x_{c,i,j}$ and $x_{c,j,i}$

REVIEWER COMMENTS

Reviewer #1 (Remarks to the Author):

The realistic case and comparison concerns are addressed. However, the introduction can be improved. Now the paragraphs in the introduction are somehow scattered and independent, making the introduction hard to follow.

1, Those explanations "However, they mainly use the blockchain to ensure consistency" in the response letter should be mentioned in the manuscript, together with literature on distributed optimization + blockchain, especially those studies in the power and energy sector.

2, Too many similar definitions are used, such as Decentralized management, Parallelizable algorithm. Please add in appendix the difference between them. What do you mean by the phrase decentralization? Everyone solving a local subproblem (which I tend to call distributed optimization)? Everyone solving a global problem on blockchain? Which decentralization is used in your method?

3, I still believe that the proposed method is similar to (or modified based on) the combined use of distributed optimization (Everyone solving a local subproblem and then iterating until convergence) and blockchain. The difference lies in steps 4-7 in fig 2, where subproblems are not solved by DES owners but by Computation parties. Please compare them in the manuscript, ideally in fig 2.

Reviewer #2 (Remarks to the Author):

While the paper is improved, the reviewer still has several critical questions:

1. Verification of DES data authenticity. TEE solutions are designed to ensure trusted computation. However, a challenge will arise when edge devices require data from DESs. How can the edge devices guarantee the authenticity of DES data, ensuring it will not be deliberately falsified? Even if TEE solutions ensure the computation will be carried out logically based on the input data, the integrity of the output will be compromised if the input data itself is flawed.

2. Security of transmitted subproblems in less secure environments. Even if TEE-backed edge devices can generate correct subproblems, these often need to be transmitted through non-secure environments to the network or stored. Attackers could tamper with the output data during this transmission phase, such as replacing or modifying the TEE-generated results. Thus, TEE solutions alone will not be able to fully address security concerns; additional measures will be needed to ensure that the correct subproblems cannot be altered, or to ensure the receivers can detect if they are falsified.

3. Ensuring consistency across miners. The Step 7 in the paper involves two groups of nodes: the blockchain miners and the computing nodes. It describes how each computing node broadcasts its

computation results to all miners, and the miners will then verify the received messages. However, a malicious computing node could potentially send computation results to only a subset of miners, leading to inconsistencies in the miners' views. This discrepancy could prevent Miner A from ensuring its perspective aligns with Miner B's, undermining the consistency across the blockchain.

Reviewer #3 (Remarks to the Author):

The authors have addressed all my comments.

Author Response of

Break down the Trilemma in Management of Distributed Energy Systems: Decentralization, Security and Privacy

Qinghan Sun, Huan Ma, Tian Zhao, Yonglin Xin, Qun Chen*
Nature Communications

RC: Reviewer Comment, **AR: Author Response,** Manuscript text

Dear reviewers,

Thank you for the valuable comments and suggestions concerning our manuscript [NCOMMS-23-37463A-B] "Break down the Trilemma in Management of Distributed Energy Systems: Decentralization, Security and Privacy". On the basis of your comments, we have further revised the introduction part and the explanations to our method. In the discussion section we also discuss potential limitations to our work. Please find our itemized responses in below and the revised manuscript in the re-submitted files. We hope that the responses and the revised manuscript have addressed all your remaining issues.

Authors' responses to Reviewer 1

The general comments from the reviewer 1:

The realistic case and comparison concerns are addressed. However, the introduction can be improved. Now the paragraphs in the introduction are somehow scattered and independent, making the introduction hard to follow.

AR: Thank you for the suggestion. We have further refined the introduction part to clarify the concept of decentralized management and two possible parts or techniques(parallelizable algorithms and blockchain) towards decentralization. Their problems are also mentioned. Please refer to the manuscript or the following responses for further details.

RC: *1. Those explanations "However, they mainly use the blockchain to ensure consistency" in the response letter should be mentioned in the manuscript, together with literature on distributed optimization + blockchain, especially those studies in the power and energy sector.*

AR: Thank you for your valuable suggestion. We now supplement those explanations in the manuscript. As for the literature on distributed optimization + blockchain in energy sectors, in our revision we have introduced 6 related works and explain their problems.

So far, several studies have realized the security risks in PA and introduced blockchain to enhance integrity of the PA-based management process[1, 2, 3, 4, 5]. They use blockchain to aggregate the decision results provided by DES participants according to different mathematical algorithms and endorse its consistency, i.e., a DES participant cannot fool other participants with different results and all participants work on the same iteration status. Such data aggregation on blockchain in plaintext compromises the privacy-preserving feature of PA. In [6], price signals are designed as the only information exchanged to protect privacy. However, the method lacks the support to DESs with diverse energy supply characteristics, and the power transmission process and related physical constraints are not considered. Apart from these concerns, dishonest individual participants may still deliberately disrupt the convergence process to hazard security.

RC: *2. Too many similar definitions are used, such as Decentralized management, Parallelizable algorithm. Please add in appendix the difference between them. What do you mean by the phrase decentralization? Everyone solving a local subproblem (which I tend to call distributed optimization)? Everyone solving a global problem on blockchain? Which decentralization is used in your method?*

AR: **Parallelizable algorithm** is one possible part, or one possible technical means of **decentralized management**. With the phrase decentralization or decentralized management, we mean that the final management decision is not made by a single party with superior position. In early researches in energy sectors, decentralized management usually refers to using PA to do distributed computing. Later after the blockchain is introduced, the scheme where every miner works on the same global problem is also called decentralized management. Since no final decision center is involved, both conforms to our definition to decentralization. Researches may also try to combine these methods to realize decentralization. The quotes of our manuscript in below explains the concepts in detail.

Our method employs both PA and blockchain, but it is not a simple superposition. Our contribution lies in the new framework to resolve the privacy and security conflicts between PA and blockchain. We do not simply require every DES owner or the blockchain to solve a local or a global problem. Instead, we design a decentralized cooperation framework with different layers and different division of labour(see Fig. 1f). The miners and the iteration process are organized based on the theory of PA or blockchain. The theories ensure effective operation in case of no manipulations and no privacy requirements. On this basis, we analyze PA and blockchain's weaknesses in certain scenarios in terms of privacy and security, and use the framework to mend the gap. The **Decentralized Computation Layer** and **Verification & Record Layer** are responsible for verifiable subproblem solutions which avoids capricious behaviors from DES owners, while **Encrypted Modelling Layer** is responsible for key generation and partial exchange and subproblem obfuscation, which protects sensitive information of the DES owners. The proper on-chain and off-chain designs ensure a secure and privacy-preserving decentralized management.

Decentralized management of multiple DESs is a possible solution. Specifically, the decision is not made by a single party with superior position. Everyone's perception is incorporated into decision-making on the expected operation status of the DESs for a certain target, such as minimization of total energy cost. ...

Blockchain is a powerful solution to realize secure decentralized management. Through smart contracts, the publicly-agreed functionalities are encoded on the blockchain and executed by miners, during which process a consensus must be formed among participants. ...

Parallelizable algorithm(PA) for mathematical optimization is another popular decentralized implementation[7, 8], but differs in the idea behind. PA-based design usually enables local computation to maximize the degree of decentralization and confidentiality[9]. Through mathematical algorithms such as distributed dual ascent[10] and Consensus+Innovation method[11], the integrated optimization problem is decomposed and every DES collects information from neighbours and solves local subproblems iteratively to reach a consensus. ...

Blockchain and PA focus on different concerns of decentralization and thus indicate respective limitations. ... So far, several studies have realized the security risks in PA and introduced blockchain to enhance integrity of the PA-based management process[1, 2, 3, 4, 5]. They use blockchain to aggregate the decision results provided by DES participants according to different mathematical algorithms and endorse its consistency, ...

In captions of Fig 1 of the manuscript, we have further clarified the definitions and differences between different decentralizations.

RC: *3. I still believe that the proposed method is similar to (or modified based on) the combined use of distributed optimization (Everyone solving a local subproblem and then iterating until convergence) and blockchain. The difference lies in steps 4-7 in fig 2, where subproblems are not solved by DES owners but by Computation parties. Please compare them in the manuscript, ideally in fig 2.*

AR: Thank you for the comments. As we have discussed in the response to Comment 1 and 2, existing researches have tried to combine the two techniques simultaneously. They let every DES participant to solve a local subproblem, aggregate and post the results on blockchain to ensure consistency of the subproblem solutions, and iterate until convergence. Their optimization framework can be summarized in Fig.1e. In Fig.1a, we show the non-cooperative case. In Fig.1b, we show that using a decision center results in security and privacy problems. In Fig.1c, we show that incorporating blockchain solves the security problem, but privacy is still compromised. In Fig.1d, we show that PA ensures privacy, but decision security is threatened. In Fig.1e, we show how blockchain and PA are combined in energy sectors currently. The privacy and security problems are not resolved:

1. **The use of blockchain does not ensure decision security.** Existing researches use the blockchain to ensure consistency. In other words, every participants share the same global information, and this global information is tamper-proof. But they neglect the potential malicious behaviors of the DES. DESs are interested parties but are still allowed to locally formulate and solve the subproblems. This leaves a loophole. For instance, a dishonest DES can formulate false local optimization subproblems that involve false costs functions about inter-DES transmission line power. Solving the false subproblems may lead to a false higher final cost and locational marginal price, and the dishonest DES benefits from this. Besides, irrational malicious participants can even prevent the convergence of the iterations. In the revised manuscript we have shown this.

Figure 1: Comparisons between different operation paradigms. **a.** The DESs operates in a non-cooperative manner. They tell the system operator the amount of electricity that they want to purchase, and the operator unconditionally fulfill all requests. **b.** The DESs cooperates through a centralized dispatch. They submit information of their equipments to the operator and wait for its operation schedule. **c.** Blockchain is employed to replace the centralized operator. **d.** PA is used for decentralization. DES owners solve local optimization subproblems, communicate with each other, and iterate to convergence. **e.** Existing works that directly combine PA and blockchain. Blockchain is used to ensure consistency in the optimization status. But dishonest DES participants can still provide falsified operation schedules and compromise security. Privacy problem remains. **f.** The proposed decentralized cooperation paradigm, where the centralized optimization problem is locally decomposed into subproblems and securely encrypted by TEE-backed edge devices to mask sensitive information. With blockchain and computation parties, they are verifiably solved to ensure robust convergence to optimality.

2. **The privacy-preserving feature of PA is compromised.** The topology of the power grid, the DES's decisions sensitivity to the price signals, and sometimes the plaintext of decision variables, may be

directly disclosed on the blockchain. Or they are not directly exposed but everyone can infer from the public available blockchain about these information. This causes privacy problems. From the private data, powerful energy producers may construct an information superiority and abuse its dominant market position to earn more profits. In the supplementary materials we have shown such unfair behaviours.

As comparison, our work is summarized in Fig.1f. We focus on resolving the conflicts between privacy and security in existing researches. The proposed new framework splits the subproblem solution process optimization into on-chain and off-chain layers. On the **Decentralized Computation Layer** and **Verification & Record Layer**, we incorporate third-party computation sources and blockchain to ensure the feasibility and optimality of subproblem solutions. The first problem is thus resolved, because the DES owners are not allowed to solve the subproblems directly and do not have a chance to cheat on the solutions. In this process, another problem arises that how to let computation parties and blockchain miners solve and verify these subproblems while preventing leakage of private information. To solve this, we use **Encrypted Modelling Layer** to break the sensitive data into encrypted slices and only the obfuscated data will be disclosed on the blockchain. Analysis of solutions to the encrypted problem may only results in random numbers and eigenvalues. Only the DES owner and its neighbours with keys can fully or partially recover the original decision variable. Therefore, the solution and verification on blockchain will not compromise the privacy of DES owners, and the second problem is overcome. Moreover, our design enhances usability for lightweight users, because the DES owners are no longer required to be equipped with a high performance computer to solve the possible sticky subproblems. Our work does not simply superpose blockchain on PA. We design a new scheme where the theories of PA and blockchain are used, but their drawbacks and contradictions are analyzed and avoided. By incorporating different participants into different data and computation layers, we ensure secure, privacy-preserving decentralized management.

These explanations are also supplemented in the revised manuscript.

Authors' responses to Reviewer 2

The general comments from the reviewer 2:

While the paper is improved, the reviewer still has several critical questions:

RC: *1. Verification of DES data authenticity. TEE solutions are designed to ensure trusted computation. However, a challenge will arise when edge devices require data from DESs. How can the edge devices guarantee the authenticity of DES data, ensuring it will not be deliberately falsified? Even if TEE solutions ensure the computation will be carried out logically based on the input data, the integrity of the output will be compromised if the input data itself is flawed.*

AR: Thank you for your comments about the data authenticity. Actually, these questions are related to identity attestation of the participants. A general solution is to use digital certificates based on asymmetric cryptography. In a digital certificate, a pair of public key and private key are involved. The public key is made public, while the private key is confidentially kept by the certificate owner. If the certificate owner wants to send a message to other people, he can use the private key to generate a digital signature of the data. Using the public key, everyone can verify whether the message is indeed sent by the certificate owner and whether the message has been distorted. In our work, the digital certificates of all DES owners are recorded on the blockchain, with each corresponding to an edge device. The TEE uses the public key to ensure that the input data has a right signature, and thus is from the 'right' DES owner.

RC: *Security of transmitted subproblems in less secure environments. Even if TEE-backed edge devices can generate correct subproblems, these often need to be transmitted through non-secure environments to the network or stored. Attackers could tamper with the output data during this transmission phase, such as replacing or modifying the TEE-generated results. Thus, TEE solutions alone will not be able to fully address security concerns; additional measures will be needed to ensure that the correct subproblems cannot be altered, or to ensure the receivers can detect if they are falsified.*

AR: Thank you for your question. To ensure information transmission security. Two aspects have to be considered: 1. The communication counterpart is exactly the one we want to communicate with. 2. The communication channel are well protected from manipulations.

The first can be guaranteed by the remote attestation services provided by TEE hardware producers. The remote attestation is a process to let a TEE hardware gain trust from a remote party, so that a remote party have confidence in the TEE hardware and the software that is running in it. Commercial TEE solutions, such as Intel's Software Guard Extensions(SGX), or ARM's Trustzone, or AMD's TEE, are all provided with this function. Apart from these solutions from well-known enterprises, many embedded computer manufacturers can set up their own attestation systems by letting the hardware generate its own digital certificate for attestation.

The second can be guaranteed by secure communication protocols, e.g., Hypertext Transfer Protocol Secure(HTTPS) and WebSocket Secure(WSS). These protocols are widely used and are important and fundamental parts of modern internet. They can help set up a tamper-proof communication channel between participants. With these internet infrastructure, the communication can be protected. In our paper, we used WSS protocol for the data exchange.

With the help of remote attestation and secure communication protocols, it is believed that both participant authenticity and the communication security are guaranteed. We assume that these techniques are mature and reliable. Discussions on their vulnerabilities are out of scope of our work.

RC: *Ensuring consistency across miners. The Step 7 in the paper involves two groups of nodes: the blockchain miners and the computing nodes. It describes how each computing node broadcasts its computation results to all miners, and the miners will then verify the received messages. However, a malicious computing node could potentially send computation results to only a subset of miners, leading to inconsistencies in the miners' views. This discrepancy could prevent Miner A from ensuring its perspective aligns with Miner B's, undermining the consistency across the blockchain.*

AR: We agree with the reviewer's concern. Even if there is no such malicious computing node, the network communication delay may result in such information discrepancies. In our code, this problem is solved by a compulsory information rebroadcast. All of the following miners are required to forward the computation parties' solutions to the leading miner. When the leading miner A collects all subproblem solutions and initiate the pBFT protocol, if the following miner B finds that miner A claims a preprepare message but it misses some subproblem solution information, it will request the corresponding information from Miner A. If Miner A refuses to provide such information, miner B may ask to overthrow A.

References

- [1] M. Foti, C. Mavromatis, and M. Vavalis, "Decentralized blockchain-based consensus for optimal power flow solutions," *Applied Energy*, vol. 283, p. 116100, 2021.

- [2] S. Wang, Z. Xu, and J. Ha, "Secure and decentralized framework for energy management of hybrid ac/dc microgrids using blockchain for randomized data," *Sustainable Cities and Society*, vol. 76, p. 103419, 2022.
- [3] G. van Leeuwen, T. AlSkaif, M. Gibescu, and W. van Sark, "An integrated blockchain-based energy management platform with bilateral trading for microgrid communities," *Applied Energy*, vol. 263, p. 114613, 2020.
- [4] Y. Wu, X. Zhang, and H. Sun, "A multi-time-scale autonomous energy trading framework within distribution networks based on blockchain," *Applied Energy*, vol. 287, p. 116560, 2021.
- [5] M. Yan, F. Teng, W. Gan, W. Yao, and J. Wen, "Blockchain for secure decentralized energy management of multi-energy system using state machine replication," *Applied Energy*, vol. 337, p. 120863, 2023.
- [6] B. Wang, S. Zhao, Y. Li, C. Wu, J. Tan, H. Li, and K. Yukita, "Design of a privacy-preserving decentralized energy trading scheme in blockchain network environment," *International Journal of Electrical Power & Energy Systems*, vol. 125, p. 106465, 2021.
- [7] A. Kargarian, J. Mohammadi, J. Guo, S. Chakrabarti, M. Barati, G. Hug, S. Kar, and R. Baldick, "Toward distributed/decentralized dc optimal power flow implementation in future electric power systems," *IEEE Transactions on Smart Grid*, vol. 9, no. 4, pp. 2574–2594, 2018.
- [8] B. Kim and R. Baldick, "A comparison of distributed optimal power flow algorithms," *IEEE Transactions on Power Systems*, vol. 15, no. 2, pp. 599–604, 2000.
- [9] D. K. Molzahn, F. Dörfler, H. Sandberg, S. H. Low, S. Chakrabarti, R. Baldick, and J. Lavaei, "A survey of distributed optimization and control algorithms for electric power systems," *IEEE Transactions on Smart Grid*, vol. 8, no. 6, pp. 2941–2962, 2017.
- [10] S. Bolognani, R. Carli, G. Cavraro, and S. Zampieri, "Distributed reactive power feedback control for voltage regulation and loss minimization," *IEEE Transactions on Automatic Control*, vol. 60, no. 4, pp. 966–981, 2015.
- [11] S. Kar, G. Hug, J. Mohammadi, and J. M. F. Moura, "Distributed state estimation and energy management in smart grids: A consensus+ innovations approach," *IEEE Journal of Selected Topics in Signal Processing*, vol. 8, no. 6, pp. 1022–1038, 2014.

REVIEWER COMMENTS

Reviewer #1 (Remarks to the Author):

Most of my comments are addressed. Fig 3 is better than before. However, I still have a question: The majority benefits and innovation seem to come from the Encrypted Modelling Layer, not the Decentralized Computation Layer and Verification & Record Layer.

"we incorporate third-party computation sources and blockchain to ensure the feasibility and optimality of subproblem solutions. The first problem is thus resolved"

Usefulness: The first problem is to defend the potential malicious behaviors of the DES, which has anything to do with third-party computation sources and blockchain?

Novelty: studies that combine distributed optimization and blockchain, such as "A distributed and robust security-constrained economic dispatch algorithm based on blockchain" "A trusted energy trading framework by marrying blockchain and optimization", can also incorporate third-party computation sources and blockchain, by letting professional computation sources be miners. Please discuss and compare.

Reviewer #2 (Remarks to the Author):

The authors have addressed the reviewer's comments.

Author Response of

Break down the Trilemma in Management of Distributed Energy Systems: Decentralization, Security and Privacy

Qinghan Sun, Huan Ma, Tian Zhao, Yonglin Xin, Qun Chen*
Nature Communications

RC: Reviewer Comment, AR: Author Response, Manuscript text

Dear reviewers,

Thank you for the valuable comments and suggestions concerning our manuscript [NCOMMS-23-37463C] "Break down the Trilemma in Management of Distributed Energy Systems: Decentralization, Security and Privacy". Please find our itemized responses in below and the revised manuscript in the re-submitted files. We hope that the responses can address all your remaining issues.

Authors' responses to Reviewer 1

General comments from Reviewer 1:

RC: *Most of my comments are addressed. Fig 3 is better than before. However, I still have a question: The majority benefits and innovation seem to come from the Encrypted Modelling Layer, not the Decentralized Computation Layer and Verification & Record Layer.*

AR: Thank you for the question. In existing researches that involve blockchain such as [1, 2], the miners are organized as a whole to solve the global optimization problems. That is, the miners collect all of the information and one miner solves the global optimization problem while others check the solution correctness. In existing researches that combine decentralized optimization and blockchain such as [3, 4, 5, 6, 7, 8], the data aggregation of distributed solution results is performed on the blockchain. Therefore, they face the risk of privacy leakage.

To protect privacy, we use the Encrypted Modelling Layer to break up the entire optimization problem into encrypted local subproblems. To ensure secure solution of these subproblems, computation sources and verifications are necessary. Due to privacy concerns of the encryption keys, these encrypted local subproblems are not reorganized as a whole and solved in an integrated manner. Instead, we incorporate the Decentralized Computation Layer and Verification & Record Layer to handle them. These subproblems are not solved by a single entity or a certain group of entities, but by any computation sources who provide solutions accepted by the blockchain. With the two layers, any third-party computation sources can join the framework, share their computing power and contribute to the management process. The optimization subproblems are allocated to the competent computation sources in a fully distributed manner. Their actions in the management process are supervised by the miners. Therefore, the Decentralized Computation Layer and Verification & Record Layer make the decentralized solution process more open, secure and efficient. In the meanwhile, privacy is protected.

RC: *"we incorporate third-party computation sources and blockchain to ensure the feasibility and optimality of subproblem solutions. The first problem is thus resolved"*

Usefulness: The first problem is to defend the potential malicious behaviors of the DES, which has anything to do with third-party computation sources and blockchain?

Novelty: studies that combine distributed optimization and blockchain, such as "A distributed and robust security-constrained economic dispatch algorithm based on blockchain" "A trusted energy trading framework by marrying blockchain and optimization", can also incorporate third-party computation sources and blockchain, by letting professional computation sources be miners. Please discuss and compare.

AR: Thank you for the questions on the usefulness and novelty of the Decentralized Computation Layer and Verification & Record Layer. Our responses are as follows:

Usefulness: In our work, the third-party computation sources and blockchain solve and verify the encrypted subproblems, i.e., they partly substitute the function that originally belong to the DESs owners to avoid their malicious behaviors. This is because in the case study of the paper we have shown that, if we let DES owners solve the optimization subproblems without verification from third parties, they may cheat on the solutions and benefit from this while increasing energy costs of other DESs. Even when they do not cheat other participants with different computing results, there is still a way for them to earn unjust profits. DESs' malicious behaviors of this kind is what we want to prevent. Since the computation parties and the miners are supervised by blockchain, the correctness of their solution to subproblems is guaranteed, and the malicious DES owners cannot cheat on the results.

Novelty: The studies can also incorporate third-party computation sources and blockchain by letting professional computation sources be miners. This is technically feasible. Most blockchains, such as Bitcoin, Ethereum and those designed for specific problems, allow this through the concept of smart contract. But considering that the blockchain consensus algorithms are usually computation-intensive enough and may suffer from performance bottlenecks[9], researchers tend to perform the user-defined computation-intensive or storage-intensive tasks in an off-chain manner[10, 11, 12], and only verify the results on chain. Our work also follows this convention and thus involve a specialized computation layer. Of course, the miners can also serve as computation parties in our framework. That is, an agent can work in two or more layers simultaneously.

We then compare with the studies. If there is an outstanding computing source, let him be the miner in [8, 13] would be alright, as the professional computation sources can solve the optimization problems efficiently. But in our work there is a difference. To protect privacy, there are a lot of encrypted DES subproblems to be verifiably computed. Besides, the participating miners may not be powerful enough to solve a lot of encrypted optimization subproblems simultaneously. In this case, we introduce the computation layer to share their computing power with the miners and the DESs. Any online computation parties can contribute to the solution process, as long as they provide correct answers accepted by miners. The incorporation of these third-party computation sources and their competition can help more robustly and efficiently solve the encrypted subproblems in a completely distributed manner, even when the miners are not capable of solving a large number of optimization subproblems. Therefore, the Decentralized Computation Layer and Verification Layer realized a secure and shared computing.

With the above discussion, we can see that the idea of letting professional computation sources be miners may be covered as a special case of the proposed framework. In the special case, the miners in Verification & Record Layer are professional computation sources and together form up the Distributed Computation Layer, i.e., the two layer coincides. In our work, the Distributed Computation Layer is not limited to miners, but is open to numerous online third-party computation sources.

Moreover, with the Encrypted Modelling Layer the Decentralized Computation Layer, the proposed framework can also be implemented on popular and well-developed public chain such as Ethereum. The existence Computation Layer reduces the computation source reliance on the blockchain platforms, e.g., reduces Ethereum gas consumption(cost).

In conclusion, the novelty of our work lies in the structured design of the privacy-preserving and secure decentralized management framework. The different function layers complete the management process and are all necessary. In the paper, we implement the framework and perform a case study to show its effectiveness in privacy-preserving and secure optimization.

In the revised manuscript, we further supplement some relating literature combining distributed optimization and blockchain, and some discussion on the different function layers as shown above. We sincerely hope that the above responses have addressed all your concerns.

References

- [1] M. K. AlAshery, Z. Yi, D. Shi, X. Lu, C. Xu, Z. Wang, and W. Qiao, "A blockchain-enabled multi-settlement quasi-ideal peer-to-peer trading framework," *IEEE Transactions on Smart Grid*, vol. 12, no. 1, pp. 885–896, 2021.
- [2] S. Chen, H. Mi, J. Ping, Z. Yan, Z. Shen, X. Liu, N. Zhang, Q. Xia, and C. Kang, "A blockchain consensus mechanism that uses proof of solution to optimize energy dispatch and trading," *Nature Energy*, vol. 7, no. 6, pp. 495–502, 2022.
- [3] E. Münsing, J. Mather, and S. Moura, "Blockchains for decentralized optimization of energy resources in microgrid networks," in *2017 IEEE Conference on Control Technology and Applications (CCTA)*, pp. 2164–2171, 2017.
- [4] D. Ogawa, K. Kobayashi, and Y. Yamashita, "Blockchain-based distributed optimization for energy management systems," in *2019 IEEE International Conference on Industrial Cyber Physical Systems (ICPS)*, pp. 706–711, 2019.
- [5] M. Foti, C. Mavromatis, and M. Vavalis, "Decentralized blockchain-based consensus for optimal power flow solutions," *Applied Energy*, vol. 283, p. 116100, 2021.
- [6] S. Wang, Z. Xu, and J. Ha, "Secure and decentralized framework for energy management of hybrid ac/dc microgrids using blockchain for randomized data," *Sustainable Cities and Society*, vol. 76, p. 103419, 2022.
- [7] G. van Leeuwen, T. AlSkaif, M. Gibescu, and W. van Sark, "An integrated blockchain-based energy management platform with bilateral trading for microgrid communities," *Applied Energy*, vol. 263, p. 114613, 2020.
- [8] S. Chen, Z. Shen, L. Zhang, Z. Yan, C. Li, N. Zhang, and J. Wu, "A trusted energy trading framework by marrying blockchain and optimization," *Advances in Applied Energy*, vol. 2, p. 100029, 2021.
- [9] Z. Zheng, S. Xie, H.-N. Dai, W. Chen, X. Chen, J. Weng, and M. Imran, "An overview on smart contracts: Challenges, advances and platforms," *Future Generation Computer Systems*, vol. 105, pp. 475–491, 2020.

- [10] P. Das, L. Eckey, T. Frassetto, D. Gens, K. Hostáková, P. Jauernig, S. Faust, and A.-R. Sadeghi, “{FastKitten}: Practical smart contracts on bitcoin,” in *28th USENIX Security Symposium (USENIX Security 19)*, pp. 801–818, 2019.
- [11] J. Eberhardt and S. Tai, “On or off the blockchain? insights on off-chaining computation and data,” in *Service-Oriented and Cloud Computing: 6th IFIP WG 2.14 European Conference, ESOC 2017, Oslo, Norway, September 27-29, 2017, Proceedings 6*, pp. 3–15, Springer, 2017.
- [12] X. Chen, J. Ji, C. Luo, W. Liao, and P. Li, “When machine learning meets blockchain: A decentralized, privacy-preserving and secure design,” in *2018 IEEE International Conference on Big Data (Big Data)*, pp. 1178–1187, 2018.
- [13] S. Chen, L. Zhang, Z. Yan, and Z. Shen, “A distributed and robust security-constrained economic dispatch algorithm based on blockchain,” *IEEE Transactions on Power Systems*, vol. 37, no. 1, pp. 691–700, 2022.

REVIEWERS' COMMENTS

Reviewer #1 (Remarks to the Author):

No further comments.

Author Response of

Break down the Trilemma in Management of Distributed Energy Systems: Decentralization, Security and Privacy

Qinghan Sun, Huan Ma, Tian Zhao, Yonglin Xin, Qun Chen*
Nature Communications

RC: *Reviewer Comment*, **AR:** *Author Response*, Manuscript text

Authors' responses to Reviewer 1

RC: *No further comments.*

AR: Thank you. We appreciate your valuable comments and suggestions.